



# 1  Biomass-burning derived particles from a wide variety of

# 2  fuels: Part 1: Properties of primary particles

Crystal D. McClure[1], Christopher Y. Lim[2,%], David H. Hagan[2], Jesse H. Kroll[2], Christopher D.
Cappa[1,3,*]
[1] Department of Civil and Environmental Engineering, University of California, Davis, CA 95616
[2] Department of Civil and Environmental Engineering, Massachusetts Institute of Technology,
Cambridge, MA, USA
[3] Atmospheric Sciences Graduate Group, University of California, Davis, CA, USA 95616
[%] Now at Department of Chemistry, University of Toronto, Ontario, Canada
[*] To whom correspondence should be addressed: cdcappa@ucdavis.edu
**ABSTRACT**
Relationships between various optical, physical, and chemical properties of biomass
combustion derived particles are characterized for particles produced from a wide range of fuels
and burn conditions. The modified combustion efficiency (MCE), commonly used to parameterize
biomass particle emissions and properties, is shown to generally have weak predictive capabilities,
especially for more efficient combustion conditions. There is, however, a strong relationship
between many intensive optical properties (e.g. single scatter albedo, Ångstrom absorption
exponent, mass absorption efficiency) and the organic aerosol-to-black carbon ([OA]/[BC]) mass
ratio over a wider range than previously considered (0.3 to $10^5$). The properties of brown carbon
(BrC, i.e. light absorbing organic carbon) also vary with [OA]/[BC]. The contribution of coating-
induced enhancements (i.e. "lensing" effects) to absorption by black carbon are shown to be
negligible for all conditions. The BC-OA mixing state varies strongly with [OA]/[BC]; the fraction
of OA that is internally mixed with BC decreases with [OA]/[BC] while the relative amount of
OA coated on BC increases. In contrast, there is little relationship between many OA bulk chemical
properties and [OA]/[BC], with the O:C and H:C atomic ratios and the relative abundance of a key
marker ion ($m/z$ = 60, linked to levoglucosan) all showing no dependence on [OA]/[BC]. In
contrast, both the organic nitrate fraction of OA and the OA volatility do depend on the [OA]/[BC].
Neither the total particle or BC-specific size distributions exhibit any clear dependence on the burn
conditions or [OA]/[BC], although there is perhaps a dependence on fuel type. Overall, our results
expand on existing knowledge to contribute new understanding of the properties of particles
emitted from biomass combustion.



## 1    Introduction

While it is understood that both open and controlled biomass combustion are major sources of
particles to the atmosphere (Andreae and Merlet, 2001), questions remain regarding the properties
of the emitted particles, their relationship with combustion conditions and fuel type, and their
atmospheric evolution. Particles emitted from biomass combustion impact the global radiation
budget and contribute to poor air quality in impacted regions. The emitted primary particles are
primarily composed of organic aerosol (OA) and black carbon (BC), in varying amounts, with
trace inorganic species (Reid et al., 2005;McMeeking et al., 2009;Levin et al., 2010). Particle
intensive properties are often compared against the modified combustion efficiency (MCE ~
$\Delta[CO_2]/(\Delta[CO]+\Delta[CO_2])$), which provides a measure of the combustion efficiency of a burn. For
example, various particle properties show some relationship with MCE, but often these
relationships are weak, especially for more efficient combustion (higher MCE, corresponding
typically to flaming conditions) (McMeeking et al., 2009;Liu et al., 2013;McMeeking et al., 2014).
Understanding the diversity in the chemical, physical, and optical properties of the emitted
particles is important for establishing the fire- or region-specific emissions and subsequent
impacts.
The emitted OA from biomass combustion is somewhat light absorbing (Kirchstetter et al.,
2004). Absorbing OA is commonly referred to as brown carbon (BrC), with properties that appear
to depend on the fuel and combustion conditions (Saleh et al., 2014;Laskin et al., 2018), which
affect particle organic composition (Jen et al., 2019). However, the properties of primary BrC
absorption and, especially, understanding of the relationships between BrC absorption and other
particle properties and burn conditions is only beginning to be unraveled. Additionally, it is
established from theory and laboratory experiments that non-absorbing coatings on black carbon
and other strongly absorbing particles can enhance the absorption (commonly referred to as the
"lensing" effect but more accurately termed here the coating-induced enhancement) (Fuller et al.,
1999;Bond et al., 2006;Lack et al., 2009;Shiraiwa et al., 2010;Cappa et al., 2012). Yet, the extent
to which coating-induced enhancements impact absorption by ambient particles or for mixed-
component particles from complex sources, such as biomass burning, remains contentious (Cappa
et al., 2012;Healy et al., 2015;Liu et al., 2015;Peng et al., 2016;Liu et al., 2017).





Here, we expand on current understanding of the relationships between various primary particle properties and burn conditions by analyzing measurements of primary biomass burning particles produced from combustion of a variety of fuel types, many of particular relevance to the western U.S.. We demonstrate that various optical properties exhibit a strong relationship with the [OA]/[BC] mass ratio, much stronger than their relationship with the MCE. We use the measurements to quantify the individual contributions of BC, BrC and from internal mixing of BC to the observed light absorption, and examine the variability in the properties of BrC specifically. We uniquely characterize the mixing state of BC and OA, and how mixing state very between individual burns and depend on the mean properties of the emitted particles. We characterize the variability of OA-specific properties, including OA volatility, bulk chemical composition (characterized by the O:C and H:C atomic ratio, and the presence of key marker ions), and, uniquely, the relative abundance of organic nitrate species. We also examine the variability in the emitted particle size distribution, both for the total particles and for the BC particles specifically. Some of our analysis serves to support and extend previously determined relationships by considering a wider range of conditions, while other aspects are unique to this study. These observations provide a foundation for understanding and interpretation of experiments on the influence of photochemical aging on biomass particle properties, discussed in a related paper (Lim et al., 2019).

## 2    Methods

All experiments were conducted during the Fire Influence on Regional to Global Environments Experiment (FIREX) lab study, which took place at the Missoula Fire Sciences Lab in Missoula, MT, USA during November, 2016. Numerous types of biomass were combusted in a large chamber (12 x 12 x 19 m) and the smoke sampled to provide information on the physical, chemical, and optical properties of the resulting smoke (i.e., particulate and gas emissions). The general fuels types combusted included (exclusively or in combination): duff, dung, excelsior, straw, litter, untreated lumber, rotten debris, woody debris, shrub, herbaceous, and canopy biomass. A complete list of fuels and types is provided in **Table S1**, with further details available on the U.S. National Oceanic and Atmospheric Administration (NOAA) data archive (https://esrl.noaa.gov/csd/projects/firex/). All data used in this publication are also available on the NOAA archive, with the processed data summarized in complementary data repository (Cappa et al., 2019a).



Both "room" and "stack" burns were conducted, although here we include results only from
stack burns. During stack burns, the smoke was mixed with background room air and funneled up
a large cylindrical stack (2 m dia. x 15 m height) where it was sampled into a high-flow transfer
line at ca. $0.27$ m$^3$/s. This flow rate corresponded to sampling approximately 10% of the stack
flow. Smoke was transferred to an adjacent room via the high-flow transfer line (residence time
ca. 2 s) where it was sub-sampled through a $PM_{2.5}$ cyclone and injected into a $0.25$ m$^3$ Teflon
photochemical reaction chamber (the mini chamber). Details on the construction and operation of
the mini chamber can be found in (Lim et al., 2019). Here, we focus exclusively on the properties
of particles sampled prior to initiation of photochemical oxidation; results of the photochemical
oxidation experiments are discussed in a series of papers (Coggon et al., 2019;Lim et al., 2019).
In brief, prior to each burn, the chamber was flushed with clean air with a relative humidity (RH)
of approximately 40%. To fill the chamber, smoke was sub-sampled from the high-flow inlet and
injected across the entire burn (typically lasting for 10-20 minutes) or until the chamber
concentration reached a maximum. A suite of instruments sampled from the mini chamber at a
flow rate of approximately 6 lpm. This flow rate varied from burn to burn due to the exact suite of
instruments sampling. Clean makeup air was being injected simultaneously from a zero air
generator to equal the air being sampled out of the chamber. The sampled smoke was diluted by a
factor of ca. seven relative to the air in the high-flow inlet. Subsequent dilution after filling was
characterized by the decay of acetonitrile (ACN). Properties of the primary particles are averaged
over the 5-10 minute period after filling but before the initiation of photochemistry.
Particle-phase instrumentation sampled alternatingly every two minutes through a
thermodenuded or ambient sample line. The thermodenuder was operated at 300 ºC with a
residence time of approximately 5 s and volatilized semi-volatile components, including those that
are internally mixed with BC. The ambient line was lined with a charcoal cloth that removed excess
gases (such as VOCs, $NO_x$, and $O_3$) that could interfere with particle-phase measurements.
Comparison of thermodenuded versus ambient particles allowed for the investigation of coating
amount and volatility. The gas-phase composition in the mini chamber was similar to that sampled
directly from the fire (Koss et al., 2018;Lim et al., 2019). Particle phase instrumentation included:
a multi-wavelength cavity-ringdown-photoacoustic absorption spectrometer (CRD-PAS) and a
photoacoustic absorption spectrometer (PASS-3) for characterization of light absorption and
extinction coefficients at 405 nm, 532 nm, and 781 nm; a high resolution aerosol mass


spectrometer (HR-ToF-AMS) for characterization of non-refractory submicron particulate matter
(NR-PM$_1$) components (i.e. OA, NO$_3$, SO$_4$, NH$_4$, Cl, K); a soot photometer AMS (SP-AMS) in
laser-only mode for characterization of refractory BC and the NR-components that are internally
mixed with BC; a single particle soot photometer (SP2) for characterization of refractory BC mass
concentrations and size distributions; and a scanning electrical mobility sizer (SEMS) for
measurement of particle mobility size distributions. Further details regarding instrument operation
and calibration are provided in the Supplemental Material and in Lim et al. (2019).

**3    Results and Discussion**
**3.1    Bulk optical property relationships**
Due to the wide variety of biomass fuels and types used during FIREX, there was a substantial
diversity in the properties of primary particles produced. Previous studies have shown both the
single scatter albedo (SSA) and wavelength-dependence of absorption (the absorption Angstrom
exponent, AAE) depend on the modified combustion efficiency (MCE) (Liu et al.,
2013;McMeeking et al., 2014;Pokhrel et al., 2017). The MCE is defined here as:
$$MCE = \frac{[CO_2]}{[CO_2]+[CO]} \tag{1}$$
The SSA is defined as:
$$SSA = \frac{b_{ext}-b_{abs}}{b_{ext}} \tag{2}$$
where $b_{ext}$ is the wavelength-specific extinction coefficient and $b_{abs}$ is the wavelength-specific
absorption coefficient. The AAE is defined as:
$$AAE = -\log(\frac{b_{abs,\lambda 1}}{b_{abs,\lambda 2}})/\log\left(\frac{\lambda 1}{\lambda 2}\right) \tag{3}$$
where $\lambda_1$ and $\lambda_2$ indicate two different wavelengths, here 405 nm and 532 nm. The MCE
characterizes the overall combustion efficiency, with values closer to unity indicating more
complete combustion. In general, higher MCE correspond to more flaming combustion conditions
while smaller MCE correspond to more smoldering conditions. We find a similar relationship
between SSA$_{405nm}$, AAE, and [OA]/[BC] with MCE as previous studies (**Figure 1**) (McMeeking





et al., 2009;Liu et al., 2013;McMeeking et al., 2014;Pokhrel et al., 2017). Specifically, the
$SSA_{405nm}$ is relatively constant and near unity for MCE < ~0.9, but above this value exhibits a
rapid decline, albeit with a substantial amount of scatter (**Figure 1**a). The AAE is also relatively
constant when MCE < 0.9, with very large values (AAE ~ 8). There is a rapid, scattered decrease
in the AAE as MCE increases further (**Figure 1**b). The relationship between [OA]/[BC] and MCE
is similar, with values generally decreasing as MCE increases but a large amount of scatter (**Figure**
**1**d). There is also a general relationship between the mass absorption coefficient referenced to BC
($MAC_{BC}$) at 405 nm and the MCE, but with similar scatter as the other properties (**Figure 1**c). The
$MAC_{BC}$ is defined as:
$MAC_{BC} = b_{abs}/[BC]$                                                    (4)
The $MAC_{BC,405nm}$ includes contributions from absorption by BC, BrC, and from coating-induced
enhancement of BC absorption. These results, along with the literature, indicate that MCE can
provide guidance as to the general magnitude of these particle properties, but that the MCE is
ultimately a fairly imprecise metric, especially for the $SSA_{405nm}$.
However, we find a very strong relationship between the $SSA_{405nm}$ and the total [OA]/[BC]
ratio (**Figure 1**e). This is consistent with the findings of Pokhrel et al. (2016), who observed
something similar but over a smaller range of [OA]/[BC]. (Similarly strong relationships are
observed for SSA values at 532 nm and 781 nm (**Figure S1**), or if the [NR-PM$_1$]/[BC] are used as
OA averages 95% of the total NR-PM$_1$ mass.) Smaller [OA]/[BC] correspond to smaller $SSA_{405nm}$
values with a sigmoidal relationship observed. (Fit parameters for all fits shown are provided in
**Table S1**.) There is similarly a very strong, sigmoidal relationship between the AAE and
MAC$_{BC,405nm}$ and [OA]/[BC] (**Figure 1**f,g). The large increase in the $MAC_{BC,405nm}$ indicates that
BrC contributes substantially to the total absorption. The contributions of coating-induced
enhancements and of BrC are discussed further in Sections 3.4.1 and 3.4.2. The larger range of
[OA]/[BC] and the greater number of individual burns considered here, compared to Pokhrel et al.
(2016), allows for determination of more robust fits. Pokhrel et al. (2017) found that the absorption
enhancement at 405 nm, determined from thermodenuder measurements, increased with
[OA]/[BC] up to [OA]/[BC] ~33 (the largest value reported), consistent with our findings.
These observations demonstrate that the optical properties of the primary particles depend on the
relative amount of OA versus BC. This is as expected because OA is generally more scattering,





compared to BC, and light absorbing OA (aka BrC) typically exhibits a much stronger wavelength
dependence than BC. Based on these relationships, we divide the individual burns into different
classes (**Table 1**). We have chosen to classify particles based on the observed $SSA_{405nm}$ values;
use of [OA]/[BC] for classification yields largely similar results, given the strong relationship
between the two. The dividing lines between classes are selected to yield six classes that span the
entire range of $SSA_{405nm}$ values, from 0.23 (Class 1) to 0.97 (Class 6), with approximately equal
numbers of individual burns in each class (ca. 8-10). Partitioning the observations into different
particle classes facilitates interpretation of the photochemical evolution of the particles, to be
discussed in future work. In addition, we find that use of the Class average properties versus MCE
generally provides more representative fits to the observations (visually apparent in **Figure 1**, and
supported by the reduced $\chi^2$ for the fits).

### 3.2    OA composition and volatility

Variability in the bulk composition of the OA is characterized by the O:C and H:C atomic

ratios and the fractional abundance ($f_x$) of two marker ions, $m/z = 44$ and $m/z = 60$. The $f_{44}$ is
complementary to O:C and larger values generally indicate a greater degree of oxygenation and
the presence of carboxylic acids. The $f_{60}$ is often taken as a marker ion for biomass burning, in
particular a signature of levoglucosan and similar molecules (Schneider et al., 2006;Alfarra et al.,
2007). The high resolution ion $C_2H_4O_2^+$ contributes to and exhibits similar behavior as $f_{60}$; the
slope for $f_{C2H4O2+}$ against $f_{60}$ is 0.98. While it is known that properties such as $f_{60}$ vary in different
biomass burning samples (Schneider et al., 2006) or between near-source intercepts of different
ambient plumes (Garofalo et al., 2019), the specific dependence on burn conditions or overall
particle composition (e.g. [OA]/[BC]) has not been systematically explored to our knowledge.

The average $f_{60} = 0.022 \pm 0.01$ (1$\sigma$). The $f_{60}$ values vary non-monotonically with [OA]/[BC],

exhibiting a slight increase from Class 1 to Class 3 and then a decrease from Class 4 to Class 6
(Figure 2a). This indicates that, while $f_{60}$ is overall a useful marker ion for biomass burning, it
cannot be used to distinguish between different burn conditions. The $f_{44}$ generally decreases with
[OA]/[BC] (Figure 2b; $r^2 = 0.33$.) However, the average $f_{44}$ values for particle Classes 2-5 differ
negligibly, suggesting that $f_{44}$ might be useful in discriminating between extreme cases (e.g. Class
1 versus Class 6), but that it is of limited general use in distinguishing between burn conditions
and fuel types. The O:C atomic ratio (average = $0.37 \pm 0.09$) exhibits similar behavior—expected

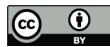



as $f_{44}$ is generally related to O:C (Aiken et al., 2008)—with a general decrease as [OA]/[BC]
increases, although a comparably weaker correlation (Figure 2c; $r^2 = 0.17$). The H:C (average =
$1.76 \pm 0.05$) exhibits a weak, positive correlation with [OA]/[BC], although the variability is slight
(Figure 2d; $r^2 = 0.27$).
The mass fraction of the OA that is composed of nitrated organics ($f_{ON-OA}$ = [ON]/[OA]) was
determined using the HR-ToF-AMS measurements and the method of Kiendler-Scharr et al.
(2016) (see the Supplemental Material for further details). The terminology nitrated organics (ON)
includes contributions from both nitro and nitrate functional groups. The fraction of measured
nitrate that was ON ($f_{ON-N}$ = [ON]/([ON]+[NO$_3^-$])) decreased with [OA]/[BC] and ranged from 0.91
(Class 1) to 0.48 (Class 6) (**Figure S2a**). The Class-specific average $f_{ON-OA}$ also decreased with
[OA]/[BC], although by a much greater extent than the $f_{ON-N}$, ranging from 6.0% (Class 1) to 0.27%
(Class 6) and (**Figure 2e**). There is a reasonably linear relationship between log($f_{ON-OA}$) and
log([OA]/[BC]) ($r^2 = 0.47$). This indicates that a larger proportion of ON species and
functionalities are produced when particles are, on average, more BC-rich. This does not reflect
differences in fuel nitrogen content as there is no relationship between fuel N and $f_{ON-OA}$ (**Figure
S2b**). Therefore, it seems that the relationship between $f_{ON-OA}$ and [OA]/[BC] is related more so to
the burn conditions than the fuel N content, although as with many other properties the relationship
with [OA]/[BC] is clearer than with the MCE (**Figure S2c**).
The OA volatility is characterized as the ratio between the OA concentration after
thermodenuding to that without thermodenuding (the mass fraction remaining, MFR$_{OA}$). The
MFR$_{OA}$ decreases as [OA]/[BC] increases (**Figure 2f**), indicating that the OA at lower [OA]/[BC]
is less volatile than the OA at higher values. This observation provides support for the proposal by
Saleh et al. (2014) that less volatile, more absorbing species are preferentially formed under
conditions where BC formation is favored, discussed further in Section 3.4.2. The relationship
between MFR$_{OA}$ and [OA]/[BC] is reasonably described by an exponential function.
**3.3   BC Mixing State**
As discussed above, the relative amounts of OA and BC vary greatly between fuel types and
combustion conditions. However, the distribution of BC and OA between particles, and how this
varies between very different burn conditions, has not been previously explored in detail to our



knowledge. The bulk average fraction of OA that is internally mixed with BC versus OA that is
externally mixed from BC is determined using the HR-ToF-AMS and SP-AMS measurements.
The HR-ToF-AMS quantifies OA independent of mixing state, whereas the SP-AMS (as operated
here) quantifies only the OA that is internally mixed with BC. The fraction of OA that is internally
mixed with BC ($f_{OA,int}$) is:

$$f_{OA,int} = \frac{[OA]_{SP-AMS}}{[OA]_{HR-ToF-AMS}} = \frac{[OA]_{int}}{[OA]_{tot}} \qquad (5)$$

where the subscript *int* indicates the OA that is internally mixed with BC and the subscript *tot*
indicates the total OA. The $f_{OA,int}$ should range from 0 to 1. Related, the SP-AMS quantified the
ratio between the OA that is internally mixed with BC and the BC concentration, referred to here
as $[OA]_{int}/[BC]$. We find that $f_{OA,int}$ decreases substantially as $[OA]/[BC]$ increases, ranging from
$f_{OA,int} = 0.4$ for Class 1 (low SSA) particles to $f_{OA,int} = 0.01$ for Class 6 (high SSA) particles (**Figure**
**3a**). The data are well-fit by a sigmoidal function. However, the amount of OA coating BC ($R_{OA-BC}$
$= [OA]_{int}/[BC]$) increases with the total $[OA]/[BC]$, also with a sigmoidal relationship (**Figure 3b**).
Thus, while a smaller fraction of the total OA is internally mixed with BC for larger total
$[OA]/[BC]$ the amount of OA that coats BC increases. Most likely this behavior reflects that BC
and OA are generated with different efficiencies in different parts of the combusting biomass. BC
is more efficiently generated from flaming combustion while OA is more efficiently generated
from smoldering combustion. These observations demonstrate that the extent to which
atmospheric models can assume that all OA is internally mixed with or externally mixed from BC
at the point of emission will depend on the combustion conditions.
**3.4   Absorption enhancement and brown carbon**
3.4.1 Coating-induced absorption enhancement
Non- or weakly-absorbing coatings on black carbon particles can theoretically increase the
absorption by BC (Fuller et al., 1999;Bond et al., 2006), an effect which has been confirmed by
laboratory experiments (Lack et al., 2009;Shiraiwa et al., 2010;Cappa et al., 2012). The extent to
which coatings on BC actually enhance absorption by BC in the atmosphere remains unclear. Some
studies indicate minor coating-induced enhancements while others indicate substantial
enhancements (Cappa et al., 2012;Healy et al., 2015;Liu et al., 2015;Peng et al., 2016;Zhang et al.,


2016;Liu et al., 2017;Cappa et al., 2019b). Understanding the nature of the coating-induced
enhancement is important for quantifying the radiative impacts of BC (Jacobson, 2001;Bond et al.,
2013). Further, these coating-induced absorption enhancements ($E_{abs.coat}$) complicate the
determination of brown carbon (BrC) absorption and the two must be separated. Here, we examine
the extent to which coatings on BC for primary biomass burning particles enhance the BC
absorption. Theoretically, the magnitude of $E_{abs,coat}$ for an individual particle depends primarily on
the coating thickness and secondarily on the size of the BC core (Bond et al., 2006; Fuller et al.,
1999). Thus, the extent to which coatings enhance BC absorption for a given situation can be
assessed through the relationship between the observed $MAC_{BC}$ and the coating-to-core mass ratio
($R_{coat-rBC}$ = [NR-PM]$_{int}$/[BC], where *int* indicates that the coating material is internally mixed with
BC). The expectation is that the $MAC_{BC}$ increases with $R_{coat-BC}$.

However, absorption by BrC can also lead to an apparent increase in the normalized absorption

with $R_{BC}$ if the BrC abundance correlates with the total coating amount. Because BrC absorbs more
strongly at shorter wavelengths, the wavelength-dependence of the $MAC_{BC}$ to $R_{BC}$ relationship can
be used to further separate the influence of coating versus BrC absorption. The $MAC_{BC}$ exhibits a
wavelength-dependent relationship with $R_{coat-rBC}$ for fresh biomass particles (405 nm, 532 nm and
781 nm) (**Figure 4a-c**). The $MAC_{BC}$ increases notably with $R_{coat-rBC}$ at 405 nm and to a lesser extent
at 532 nm. At 781 nm the $MAC_{BC}$ is essentially independent of $R_{coat-rBC}$ up to $R_{coat-rBC}$ values as
large as 10, but does exhibit some increase at $R_{coat-rBC}$ > 10. However, this is most likely a result
of absorption by OA at 781 nm and not indicative of an increase in the coating-induced
enhancement, discussed further below. The wavelength dependence provides clear evidence of
BrC absorption at shorter wavelengths.

That the $MAC_{BC}$ at 781 nm is nearly independent of $R_{coat-rBC}$ up to such large $R_{coat-rBC}$ values

indicates that there is a negligible coating-induced enhancement for the primary biomass particles.
Our observations are consistent with McMeeking et al. (2014), who also investigated the
relationship between the $MAC_{BC}$ and $R_{coat-rBC}$ for a primary biomass particles from multiple fuel
types. Most likely, this lack of a substantial coating-induced enhancement results from a non-even
distribution of non-BC mass across the population of BC particles (Fierce et al., 2016;Liu et al.,
2017) and from the morphology of BC-containing particles not conforming to an idealized core-
shell structure (Adachi et al., 2010). The influence of photochemical aging on the coating-induced
enhancement will be examined in future work.



The relationship between $MAC_{BC}$ and the coating amount ($R_{coat\text{-}rBC}$) can be contrasted with the
relationship between $MAC_{BC}$ and the total [OA]/[BC]. At all three wavelengths the $MAC_{BC}$ exhibit
strong, sigmoidal relationships with [OA]/[BC] (**Figure 4d-f**). That $MAC_{BC,781nm}$ exhibits such a
clear relationship with [OA]/[BC] suggests that even the small apparent coating-induced
enhancement, implied above from the very weak with $R_{coat\text{-}rBC}$, is largely driven by absorption by
BrC rather than from the impact of coating on BC. Pokhrel et al. (2017) found that the absorption
enhancement, determined from thermodenuder measurements, increased notably with [OA]/[BC]
up to [OA]/[BC] ~33 at 405 nm (the largest value reported by them), but by much less at 660 nm,
consistent with our findings.
The observations allow for determination of wavelength-dependent $MAC_{BC}$ values for pure BC
($MAC_{BC,pure}$) for each wavelength by extrapolation of the $MAC_{BC}$ versus [OA]/[rBC] ratio to zero
using sigmoid fits. Since the $R_{coat\text{-}rBC}$ correlates reasonably with [OA]/[BC] (**Figure 3**b),
extrapolation against [OA]/[BC] to zero effectively removes both contributions from BrC and any
coating-induced enhancement. The derived $MAC_{BC,pure}$ values are 11.8 m$^2$ g$^{-1}$ at 405 nm, 8.8 m$^2$
g$^{-1}$ at 532 nm and 5.5 m$^2$ g$^{-1}$ at 781 nm, with estimated fit-based uncertainties of ~10%. The
absolute uncertainties on the $MAC_{BC,pure}$ are primarily dependent on the uncertainty in the $b_{abs}$ and
[rBC] measurements, and are ~35%. The derived $MAC_{BC}$ values are very similar to those recently
reported by Forestieri et al. (2018) for fresh BC particles: $MAC_{BC,pure}$ = 11.9 m$^2$ g$^{-1}$ at 405 nm and
8.8 m$^2$ g$^{-1}$ at 532 nm, with an extrapolated value at 781 nm of 5.7 m$^2$ g$^{-1}$. The value at 532 nm is
somewhat higher than that suggested by Bond and Bergstrom (2006) (7.75 m$^2$ g$^{-1}$ at 532 nm). Our
derived $MAC_{BC,pure}$ values yield an $AAE$ = 1.17, determined from a fit to the three wavelengths. An
AAE close to unity indicates absorption is dominated by BC, as expected.
## 3.4.2 Primary brown carbon absorption
The absorption due to brown carbon is determined by difference as:
$$b_{abs,BrC} = b_{abs,obs} - MAC_{BC,pure} \cdot [BC] \cdot E_{abs,coat} \qquad (6)$$
where $b_{abs,BrC}$ is the absorption due to BrC specifically. Importantly, the use of study-specific
$MAC_{BC,pure}$ values serves to reduce systematic biases in the $b_{abs,BrC}$, compared to direct use of
literature $MAC_{BC,pure}$ values. Assuming $E_{abs,coat}$ = 1 provides an upper limit on the BrC absorption,
which we note is likely most appropriate for the particles sampled here, as discussed in the previous


section. Therefore, we use the upper-limit values throughout the analysis that follows, unless
otherwise stated.  However, a lower limit for BrC absorption can be determined at 405 nm and 532
nm assuming that all of the enhancement at 781 nm results from coatings and not from BrC. The
resulting $E_{abs,obs}$ (= $MAC_{BC,obs}/MAC_{BC,pure}$) at 781 nm averages 1.19 for $R_{BC\text{-}coat}$ < 10. Using $E_{abs,coat}$
= 1.19 in Eqn. 7 yields a lower limit for the BrC absorption at the two shorter wavelengths,
appropriate since $E_{abs,coat}$ generally has only a small wavelength dependence. A fit to the coating-
corrected (lower-limit) versus upper-limit $b_{abs,BrC}$ yields a slope of 0.97 at 405 nm and 0.88 at 532
nm (**Figure S3**). The smaller difference at 405 nm results from the fractional contribution of BrC
to the total absorption being larger at this wavelength.
Brown carbon-specific mass absorption coefficients ($MAC_{BrC}$) are determined as the ratio
between $b_{abs,BrC}$ and the total OA concentration:
$$MAC_{BrC} = \frac{b_{abs,BrC}}{[OA]} \qquad (7)$$
The $MAC_{BrC}$ values from Eqn. **7** are bulk-average values, and do not account for different
molecules and classes of molecules likely having different absorptivities. Uncertainties in the
$MAC_{BrC}$ values are determined by error propagation. Similarly, an AAE value for just the brown
carbon ($AAE_{BrC}$) can be calculated using wavelength pairs as:
$$AAE_{BrC} = -\log(\frac{b_{abs,BrC,\lambda_1}}{b_{abs,BrC,\lambda_2}})/\log\left(\frac{\lambda_1}{\lambda_2}\right); \qquad (8)$$
The geometric averages of the $MAC_{BrC}$ values are $0.76^{+0.65}_{-0.35}$ m$^2$ g$^{-1}$, $0.21^{+0.36}_{-0.13}$ m$^2$ g$^{-1}$, $0.056^{+0.15}_{-0.04}$
m$^2$ g$^{-1}$ at 405 nm, 532 nm and 781 nm, with uncertainties the 1σ burn-to-burn variability. The
$MAC_{BrC}$ values vary between classes, generally increasing as the [OA]/[BC] ratio decreases at all
wavelengths (shown for 405 nm in **Figure 5**a). For example, the average $MAC_{405nm}$ = 2.3 ± 1 m$^2$
g$^{-1}$ for Class 1 and 0.35 ± 0.09 m$^2$ g$^{-1}$ for Class 6. Although the uncertainties on the derived $MAC_{BrC}$
increase substantially as [OA]/[BC] decreases—because BrC absorption contributes to a smaller
extent at longer wavelengths—the observations nonetheless indicate that the BrC absorptivity
depends on the combustion conditions. The relationship at 405 nm is well-described by a sigmoidal
function in log-log space, with limiting values of 0.35 m$^2$ g$^{-1}$ at large [OA]/[BC] and 11.2 m$^2$ g$^{-1}$
at small [OA]/[BC]. That the extrapolated zero [OA]/[BC] limit for $MAC_{BrC}$ is similar to pure BC
suggests an evolution of BrC towards having properties similar to BC when the overall [OA]




content is small. Such behavior is consistent with Saleh et al. (2018), who argue that there is a
continuum of BrC properties that depends on the combustion conditions, as demonstrated in that
study for low-temperature benzene and toluene combustion. The range of the observed $MAC_{BrC}$
values here encompass many previous measurements, summarized in **Table S3**. This likely reflects
the wide diversity of fuel types and burn conditions considered here, as exemplified by the very
large range of [OA]/[BC].
Estimated values of the imaginary component of the refractive index for BrC ($k_{BrC}$) are determined
from Mie theory via optical closure (Zhang et al., 2016), assuming a real part of the refractive
index of 1.5 and a particle diameter of 150 nm, a typical value for these experiments. Imaginary
RI values are of use in atmospheric models for calculation of BrC absorption. There is a linear
relationship between $MAC_{BrC}$ and $k_{BrC}$ (**Figure S4**a). Thus, the $k_{BrC}$ exhibits a similar correlation
with [OA]/[BC] as does the $MAC_{BrC}$ (**Figure 5**a).
The wavelength-dependence of absorption, i.e. the $AAE_{405-532}$, also varies with [OA]/[BC], in this
case with a positive relationship between the two (**Figure 5**b). The relationship is reasonably
described by a sigmoidal function. This implies that, while the $MAC_{BrC}$ varies inversely with
[OA]/[BC] at all wavelengths, the exact variation is wavelength dependent. The $AAE_{405-532}$
relationship with [OA]/[BC] is well-described by a sigmoidal function (versus log([OA]/[BC]),
with limiting values of 10.4 at large [OA]/[BC] and 1.3 at small [OA]/[BC]. The wavelength-
dependence of the $k_{BrC}$ ($w_{BrC}$) are also calculated, to facilitate comparison with the literature, as:
$$w_{BrC} = -\log\left(\frac{k_{BrC,\lambda 1}}{k_{BrC,\lambda 2}}\right) / \log(\frac{\lambda 1}{\lambda 2}) \qquad (9)$$
The $w_{BrC}$ exhibit a similar dependence on [OA]/[BC] as the $AAE_{BrC}$, as the $w_{BrC}$ and $AAE_{BrC}$ are
linearly related, albeit with some scatter (**Figure S4**b; $r^2 = 0.97$).
Our observations support the results of Saleh et al. (2014), who also found a relationship between
the $k_{BrC,405nm}$ and [OA]/[BC]. However, our analysis substantially extends the range of [OA]/[BC]
values investigated in that work (they considered [OA]/[BC] from only ca. 2 to 170). In the overlap
region between our two studies the $k_{BrC,405nm}$ agree reasonably well over the range 2 < [OA]/[BC]
< 50, but the $k_{BrC,405nm}$ from Saleh et al. (2014) are smaller than observed here above [OA]/[BC] =
50. Importantly, our results demonstrate that the linear fit suggested by Saleh et al. (2014) for
$MAC_{BrC}$ is only appropriate over the range of values they considered and that a sigmoidal provides



for a more robust relationship over a wider range of [OA]/[BC]. Related, the wider range of
[OA]/[BC] enables more robust determination of the functional dependence of the wavelength-
dependence of absorption ($w_{BrC}$), with overall larger $w_{BrC}$ values and a larger plateau at high
[OA]/[BC] compared to the fit by Saleh et al. (2014).
The $MAC_{BrC}$ values also correlate with the nitrated organic fraction of OA, the latter of which, as
noted above, also correlates with the [OA]/[BC] (**Figure 6**a). This observation suggests that
organic nitrate and nitro functionalities may be at least somewhat responsible for the increase in
absorption. Laskin et al. (2018) performed offline molecular level analyses of primary OA
collected during FIREX. They found that nitroaromatics and N-containing polycyclic aromatic
hydrocarbons (PAHs) contribute notably to the total light absorption by BrC, although there are
many non-N-containing species that also contribute to BrC absorption. The variability between
particle Classes is consistent with the results of Lin et al. (2016), which show that the abundance
of N-containing chromophores varies between particles produced from different biomass fuels.
Additionally Mohr et al. (2013) observed a relationship between the concentration of nitrated
phenols and short-wavelength absorption by BrC, although it is possible that for their
measurements these species were produced from chemical processing, as opposed to being directly
emitted. Altogether, our results provide support for the idea that nitrated organic functionalities
are an important contributor to BrC absorption. However, it is very likely that other functional
groups also contribute to the total absorption.
The $MAC_{BrC,405nm}$ exhibits an inverse correlation with the $f_{60}/f_{44}$ ratio of the OA, although there is
substantial scatter in the $f_{60}/f_{44}$ ratio for a given particle class (**Figure 6**b). (The $f_{44}$ and $f_{60}$ have no
discernable relationship.) The observed $MAC_{BrC,405nm}$ relationship with $f_{60}/f_{44}$ is opposite that
reported by Lack et al. (2013) for ambient measurements of particles a biomass burning plume,
who find a reasonable positive correlation. This difference in behavior results from our sampling
primary particles directly—thereby focusing on the inherent variability in the properties of the
emitted particles—while Lack et al. (2013) sampled ambient particles. For ambient sampling, the
observed relationship will be sensitive to mixing of biomass burning particles with background or
aged biomass particles, which are known to have a smaller $f_{60}$ (Cubison et al., 2011). Thus, the
relationship observed by Lack et al. (2013) can best be viewed as a mixing line between the fresh
primary particles (having large $MAC_{BrC,405nm}$ and large $f_{60}/f_{44}$) and background or aged biomass


particles (having small $MAC_{BrC,405nm}$ and small $f_{60}/f_{44}$), rather than providing information on the
inherent variability in the absorptivity of the fresh particles.

## 3.5   Size distributions

Total particle mobility size distributions and BC-only size distributions were measured (**Figure**
**7**). Primary particle size distributions are important parameters specified in regional and global
models. The number-weighted and volume-weighted size distributionare generally described by
either one or two log-normal modes for individual burns; a two-mode fit provides a more robust
solution across all modes. The mass-weighted BC size distributions are similarly described by one
or two log-normal modes. A fit to the average number-weighted distribution across all particle
classes yields geometric median diameters ($d_{p,N}$) and widths ($\sigma_g$) of 60.3 nm and 1.76, respectively,
for the smaller mode and 153 nm and 1.64 for the larger mode (**Figure 8**). The amplitude of the
smaller mode is 4.6 times the larger mode. A single mode fit yields $d_{p,N}$ = 68 nm and $\chi_g$ = 1.93,
although the fit is poorer. Mann et al. (2014) report $d_{p,N}$ values used by a variety of global models
for biofuels. The models tend to use either 80 nm or 150 nm, although a few use other values (30
nm, 60 nm, 100 nm). Those using 80 nm typically use $\sigma_g$ = 1.80 while those using 150 nm typically
use $\sigma_g$ = 1.59, although there are exceptions. Our observations indicate that use of a bimodal
distribution within models would be more representative, but that a single mode can do acceptably.
We find that the volume-weighted distribution calculated from a single-mode fit to the number-
weighted distribution is similar to the observed volume-weighted distribution (**Figure 8**). Thus,
the use of a single-mode to represent biomass burning size distributions thus appears acceptable,
so long as the appropriate parameters are used. In this context, the widths of the distribution used
by the various global models appear somewhat too small. However, we note that the microphysics
occurring in the fresh smoke sampled here, which will govern the size distributions, may differ
from that in atmospheric plumes.
The average BC-specific mass-weighted size distribution mode is at 148 nm (**Figure 8**). A
bimodal fit yields values for the mass median diameter ($d_{p,M}$) and $\sigma_g$ of 137.2 nm and 1.62,
respectively, for the smaller mode and 197.1 nm and 1.24 for the larger mode, with most of the
mass contained in the smaller mode. May et al. (2014) report $d_{p,M}$ from laboratory biomass
combustion ranging from 140-190 nm, averaging 170 nm. Their average is somewhat larger than



ours, likely reflecting differences in the exact fuels sampled. The mode diameter for the BC-
specific distribution is especially smaller than observed for biomass burning particles from some
ambient observations, which tend to give values closer to 200 nm (Schwarz et al., 2008;Kondo et
al., 2011;Sahu et al., 2012;May et al., 2014;Cappa et al., 2019b). This difference between lab and
field observations was also noted by May et al. (2014). We speculate that the influence of
coagulation may be suppressed in our experiments relative to what occurs in the atmosphere due
to slower overall dilution, leading to smaller BC size distributions. To the extent this is the reason
for the difference, the total particle distributions would also be biased towards too small particles,
compared to the atmosphere. However, there is no relationship between $d_{p,N}$ and the total particle
number concentration for our experiments. Formation of secondary aerosol in the near-field of a
sampled ambient plume could also contribute to this difference.

There is substantial variability between individual burns within a given particle Class in terms

of the shape of the size distributions (**Figure 7**). This variability is most evident for Class 1, 2 and
5, but present for all Classes. Nonetheless, the number-weighted mean diameter ($d_{p,N,mean}$) appears
to decrease somewhat with MCE (**Figure 9**; $r^2 = 0.38$). However, the relationship is largely driven
by the Class 6 particles, which generally have lower MCE values, having larger $d_{p,N,mean}$ values. A
lack of any particularly clear relationship is consistent with Hosseini et al. (2010), who observed
the $d_{p,N,mean}$ to exhibit a complex relationship with combustion conditions. The $d_{p,N,mean}$ varies non-
monotonically with [OA]/[BC], with particle size first decreasing slightly as [OA]/[BC] increases
(from Class 1 to Class 3) and then increasing with further increases in [OA]/[BC] (from Class 4 to
Class 6) (**Figure 9**). This is despite the notable burn-to-burn variability. It is important to note that
the mobility-based size is particle shape-dependent; very BC-rich particles are more likely to have
non-spherical shapes and thus have larger mobility diameters. This could explain the minimum in
$d_{p,N}$ around Class 3 particles, for which [OA]/[BC] = 10.

Some of the variability within a class appears related to the presence of different fuel types

within a class. Number-weighted and BC-specific mass-weighted size distributions by fuel type
are shown in **Figure 10**. For the number-weighted distributions, leaf litter and rotten logs exhibit
the greatest variability between different burns, although we note that multiple burns were not
performed for all fuels. The shapes of the leaf litter, peat and "other" fuel types differ most notably
from the other fuel types, with the presence of more than one mode more apparent. (The "other"
category here includes non-traditional biofuels, specifically building materials and excelsior.) For


the BC-specific size distributions, the litter, canopy, and duff exhibited the greatest intra-fuel
variability. For most fuels, the BC-specific distribution peaks around 150 nm, as noted above.
However, for a subset of burns (eight of them) the BC-specific distribution peaks around 100 nm
(**Figure 10**). These small BC-mode distributions occur for the OA-rich particle classes 4, 5 and 6
(**Figure 7**), although there is no clear pattern to their occurrence.

## 478    4    Conclusions and Implications

Measurements of primary particles produced from combustion of a variety of biomass fuel
types indicate the optical, physical, and chemical properties of the emitted particles exhibit wide
variability. We show that variability in many optical properties (e.g. single scatter albedo,
wavelength dependence of absorption, mass absorptivity of black and brown carbon) is directly
linked to the [OA]/[BC] ratio of the emitted particles; the relationships with [OA]/[BC] are much
stronger than with the commonly used modified combustion efficiency, and mathematical
relationships between the various properties are determined. However, the absorption
enhancement due to coating of BC (the so-called "lensing" effect) is shown to be negligible and
essentially independent of the amount of coating up to large coating-to-BC mass ratios. The brown
carbon mass absorptivity correlates with the nitrated organic fraction of OA, suggesting that
nitrated organic species contribute to BrC absorption. Many bulk chemical properties (i.e. O:C,
H:C, and the relative concentrations of key marker ions such as $f_{60}$) exhibit limited dependence on
the burn conditions and the [OA]/[BC] ratio. However, both the OA volatility and nitrated organic
fraction of OA decrease with [OA]/[BC]. The fraction of OA that is internally mixed with BC was
shown to decrease strongly with the [OA]/[BC] ratio, from nearly all OA being internally mixed
with BC when the particles are overall BC-rich to only a few percent of OA being mixed with BC
when OA dominates. Yet, the relative amount of OA coating the BC increases with [OA]/[BC];
that is, when more of the OA is externally mixed from BC those particles that do contain BC
nonetheless have thicker OA coatings. The observed total particle size distributions are reasonably
well described by a single log-normal mode, but are better fit using a bimodal distribution. The
BC-specific size distributions are similarly best fit using a bimodal distribution, although a single
mode provides a reasonable representation. The dependence of the geometric median mobility
diameter on the burn conditions or particle state (i.e. the [OA]/[BC]) is complicated by the mobility
diameter being sensitive to variations in particle shape, which depend on the [OA]/[BC] ratio.
Overall, these results expand on previous observations of primary biomass burning particle


properties, considering a wider range of [OA]/[BC] and associated properties. Further, they
provide a foundation for understanding the post-emission evolution of biomass burning smoke due
to photochemical oxidation as discussed in Lim et al. (2019).
**5    Data Availability**
All data are available from the NOAA FIREX-AQ data repository
(https://esrl.noaa.gov/csd/projects/firex/firelab/). This includes a summary of the fuel types used
for each burn and the measurement time-series for each burn. The primary particle averages used
in this work are additionally collected in the UC DASH data repository (Cappa et al., 2019a).
**6    Author Contributions**
CDC and JHK designed the experiments. CDC, CYL, and DHH carried out the measurements
and data processing. CDC, CDM, and CYL analyzed data. CDC and CDM wrote the manuscript,
with contributions from all co-authors.
**7    Acknowledgements**
This work was supported by the National Oceanic and Atmospheric Administration
Atmospheric Chemistry, Carbon Cycle & Climate Program, awards NA16OAR4310111 and
NA16OAR4310112. CYL was additionally supported by the National Science Foundation
Graduate Research Fellowship Program. The entire FIREX team, especially Bob Yokelson and
Jim Roberts and the staff of the Missoula Fire Sciences Laboratory, are acknowledged for their
assistance. Putting together the community inlet was a community effort—thank you to all who
contributed. Shuka Schwarz and Gavin McMeeking are also thanked for their assistance with the
SP2.



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




## 9   Tables


**Table 1.** Fuels by particle Class.

| Class | Fuel | SSA range | [OA]/[BC] range |
|-------|------|-----------|-----------------|
| Class 1 | Chaparral, canopy, litter (pine), building materials, excelsior | 0.23-0.43 | 0.3-2.4 |
| Class 2 | Manzanita, Sage, litter (fir) | 0.43-0.60 | 1.5-4.1 |
| Type 3 | Pine, fir, litter, canopy, juniper | 0.60-0.74 | 6.6-20 |
| Class 4 | Pine, fir, canopy, rotten log, ceonothos | 0.74-0.87 | 8.3-55 |
| Class 5 | Canopy (pine), rice, bear grass, duff | 0.87-0.93 | 31-143 |
| Class 6 | Rotten log, duff, peat, dung | 0.93-1.00 | $431\text{-}10^5$ |



**10 Figures**

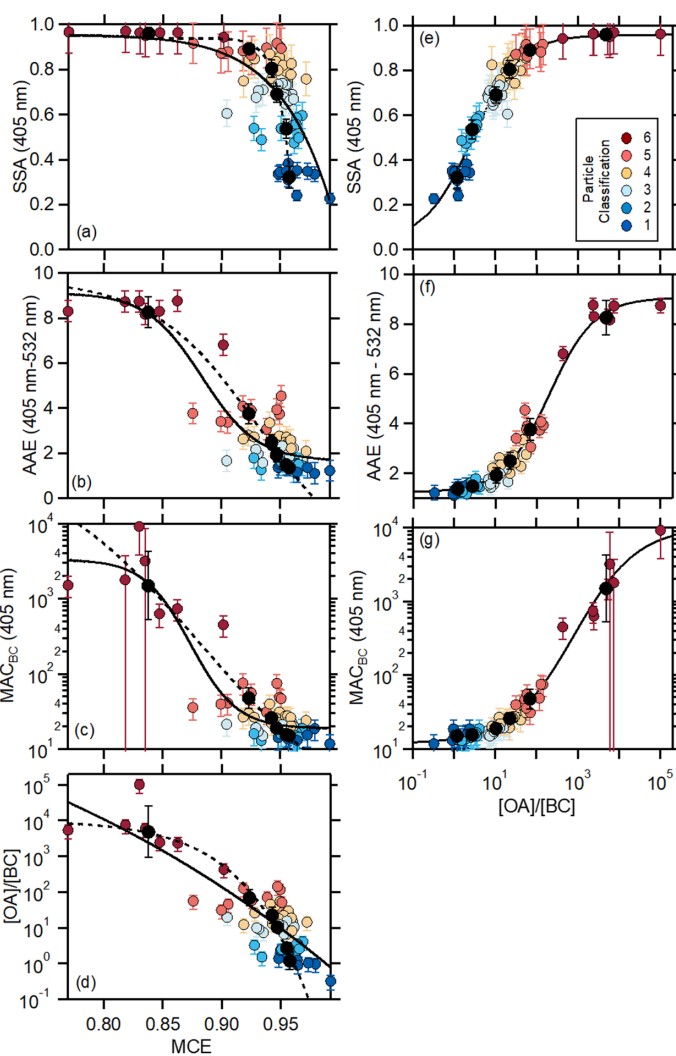


**Figure 1.** (left panels) Relationship between (a) the SSA$_{405nm}$, (b) the AAE$_{405-532}$, (c) the MAC$_{BC}$, and (d) the [OA]/[BC] mass ratio and the modified combustion efficiency, MCE. Results for individual burns are shown as points colored by the particle Class, and Class average values are shown as black circles. Uncertainties on the Class averages are 1σ based on measurement variability and uncertainties on for the individual burns are from error propagation of measurement uncertainties. The solid black lines are fits to the individual burns (colored points) while the dashed black lines are fits to the Class averages (Table S2). (right panels) Relationship between (e) the SSA$_{405nm}$, (f) the AAE$_{405-532}$, and (g) the MAC$_{BC}$ on the [OA]/[BC] mass ratio. The solid black lines here are sigmoidal fits to the individual burns. Fits to the Class averages are similar.



856

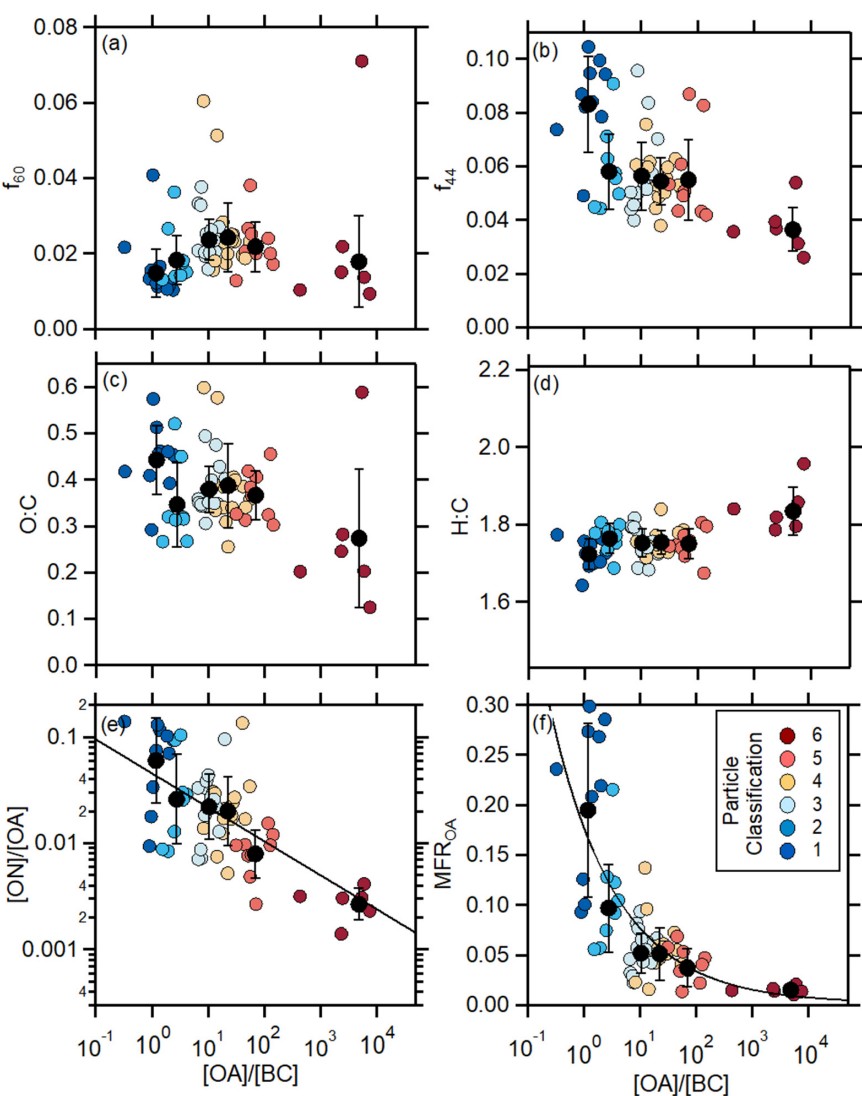

857

**Figure 2**. Dependence of (a) $f_{60}$, (b) $f_{44}$, (c) O:C, (d) H:C, (e) the nitrated organic fraction of OA, $f_{ON\text{-}OA}$, and (f) the OA volatility, characterized as the mass fraction remaining after heating. Results for individual burns are shown as points colored by the particle Class, and Class average values are shown as black circles. Uncertainties on the Class averages are 1σ based on measurement variability. For $f_{ON\text{-}OA}$ and $MFR_{OA}$, fits to the observations are shown (see text).

863

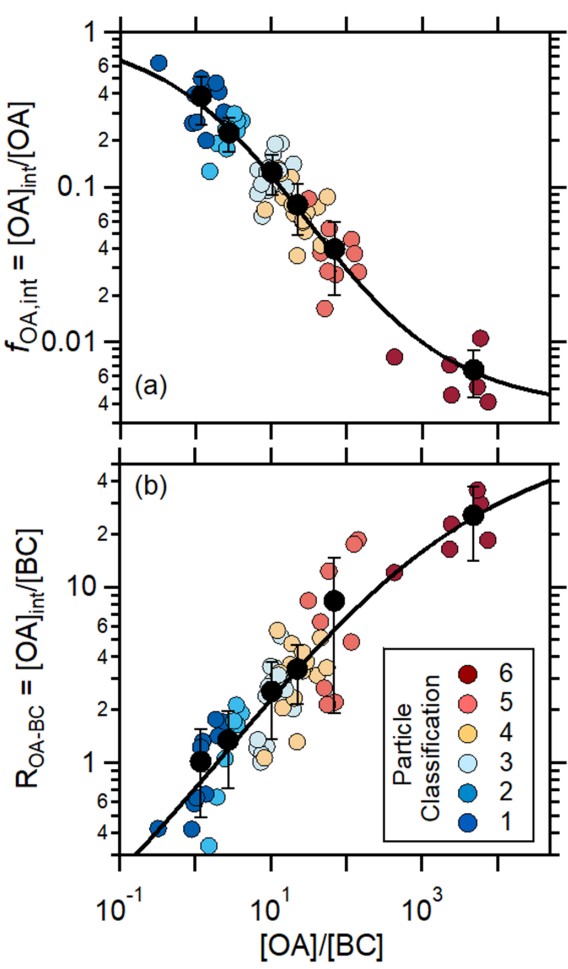

864

**Figure 3.** Relationship between (a) the fraction of OA that is internally mixed with BC, $f_{OA,int}$ and (b) the OA-to-BC mass ratio for only the internally mixed OA, and the total [OA]/[BC] mass ratio. Results for individual burns are shown as points colored by the particle Class, and Class average values are shown as black circles. Uncertainties on the Class averages are 1σ based on measurement variability. Black lines are sigmoidal fits to the data, in log-log space.

870



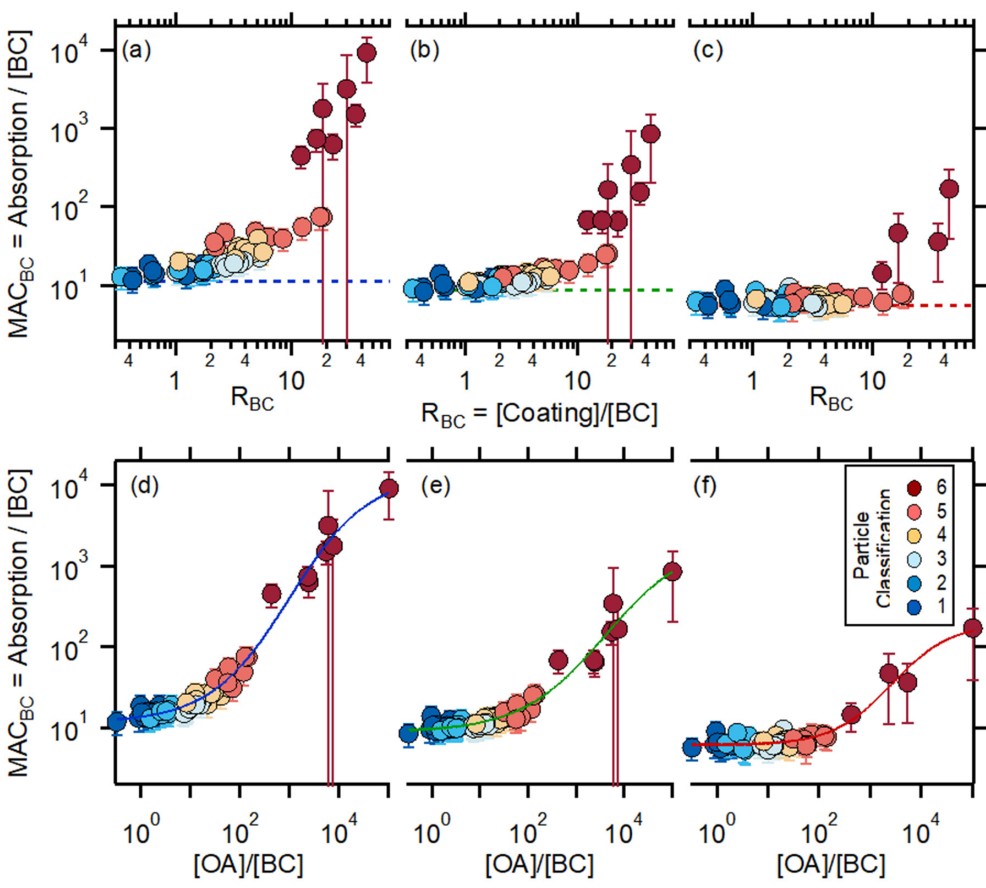

871

**Figure 4.** (Top Panels) The relationship between the wavelength-dependent $MAC_{BC}$ and the coating-to-BC mass ratio for (a) 405 nm, (b) 532 nm and (c) 781 nm. The horizontal dashed lines show the derived $MAC_{BC,pure}$ values. (Bottom Panels) The relationship between the wavelength-dependent $MAC_{BC}$ and the total [OA]/[BC] mass ratio for (a) 405 nm, (b) 532 nm and (c) 781 nm. The lines are sigmoidal fits. Uncertainties for the individual burns are determined from error propagation.

878



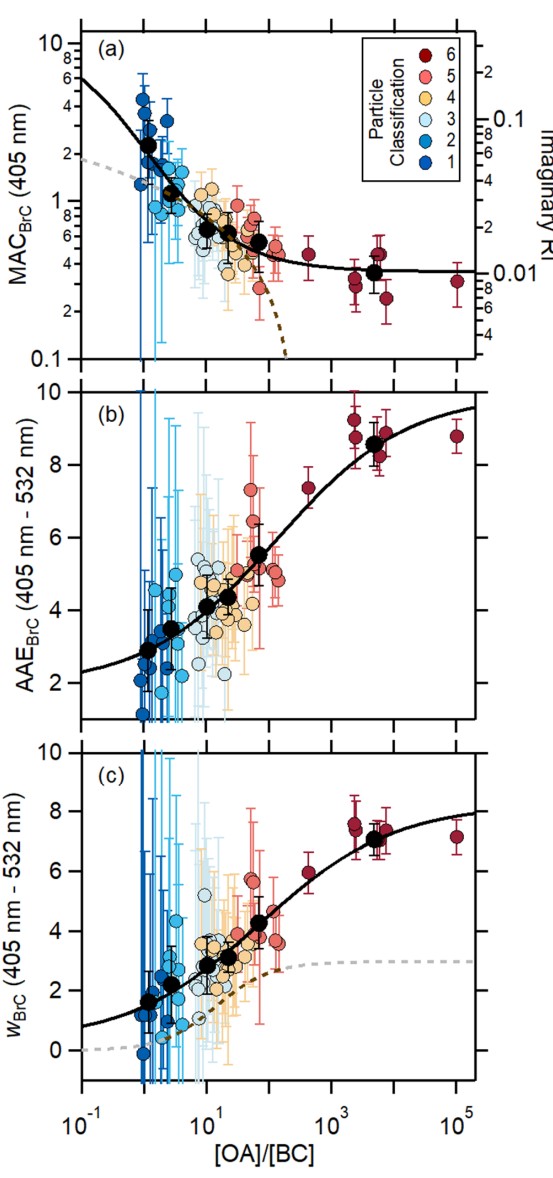

879

**Figure 5.** Relationship between (a) $MAC_{BrC,405nm}$, (b) $AAE_{BrC,405-532}$, and (c) $w_{BrC,405-532}$ and the
[OA]/[BC] mass ratio. The solid lines are sigmoidal fits to the observations, against
log([OA]/[BC]). The dashed lines are based on the parameterization of Saleh et al. (2014), with
the brown color indicating the measuring range in that study and the gray color extrapolated.
Results for individual burns are shown as points colored by the particle Class, and Class average
values are shown as black circles. Uncertainties on the Class averages are 1σ based on
measurement variability. Uncertainties for the individual burns are determined from error
propagation.



888

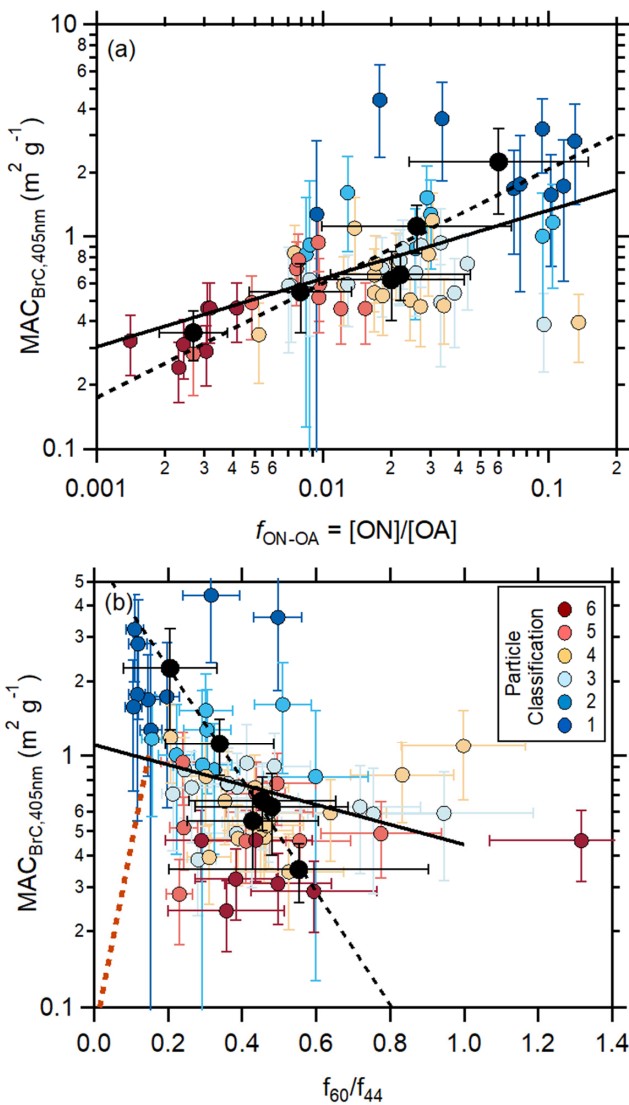

889

**Figure 6.** Relationship between the $MAC_{BrC,405nm}$ and (a) the nitrated organic fraction of total organic aerosol, $f_{ON-OA}$, and (b) the $f_{60}/f_{44}$ ion ratio for organic aerosol. Results for individual burns are shown as points colored by the particle Class, and Class average values are shown as black circles. Uncertainties on the Class averages are $1\sigma$ based on measurement variability. Uncertainties for the individual burns are determined from error propagation. Solid black lines are fits to all burns and dashed black lines are fits to the Class averages. The dashed brown line in panel (b) is the relationship reported by Lack et al. (2013) for ambient particles in a biomass burning plume.

897

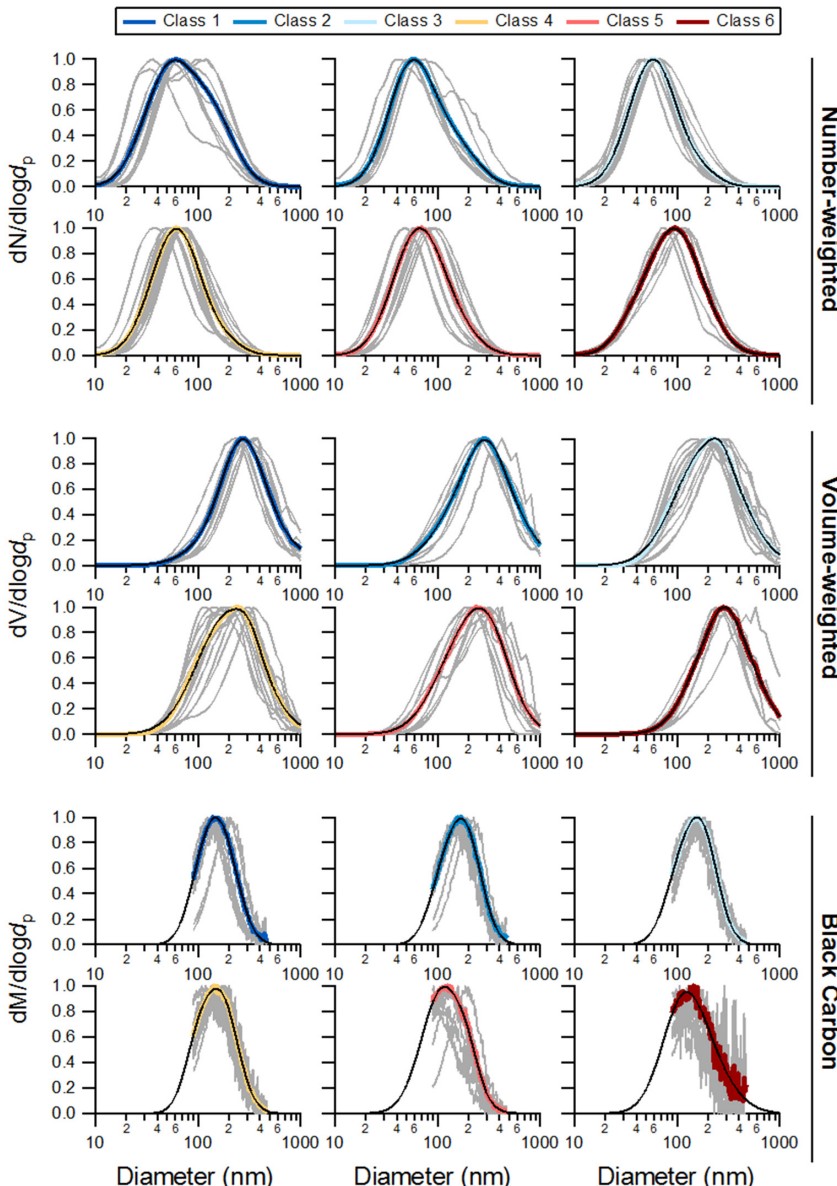

898

**Figure 7.** Class-specific total particle number-weighted (top) and volume-weighted (middle) mobility size distributions, and the BC-only mass-weighted (bottom) size distribution. Individual burns are shown in gray and class averages are shown as colors. Bimodal log-normal fits are thin black lines. Note that the number-weighted and volume-weighted distributions are graphed versus mobility diameter and the BC mass-weighted distribution against the BC volume equivalent diameter.

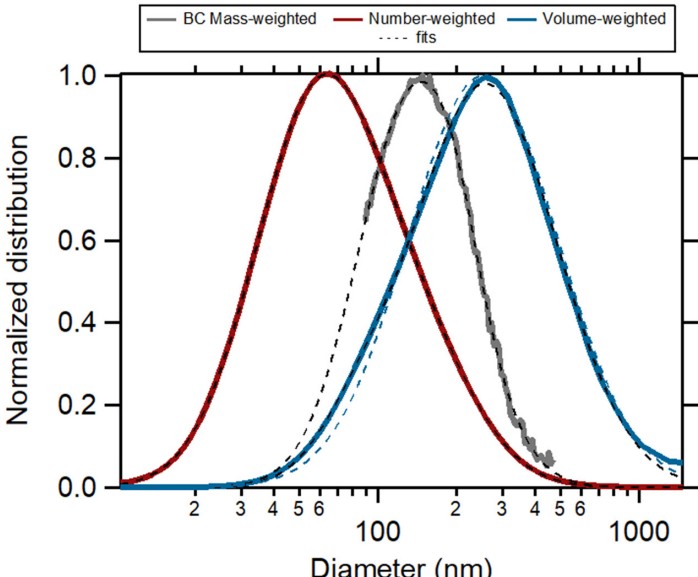

905

**Figure 8.** Average total particle number-weighted (red) and volume-weighted (blue) size
distributions and the BC-specific mass-weighted size distributions. Black dashed lines are bimodal
log-normal fits. The dashed blue line is the total particle volume-weighted distribution calculated
from a single-mode fit to the number-weighted distribution. Note that the number-weighted and
volume-weighted distributions are graphed versus mobility diameter and the BC mass-weighted
distribution against the BC volume equivalent diameter.




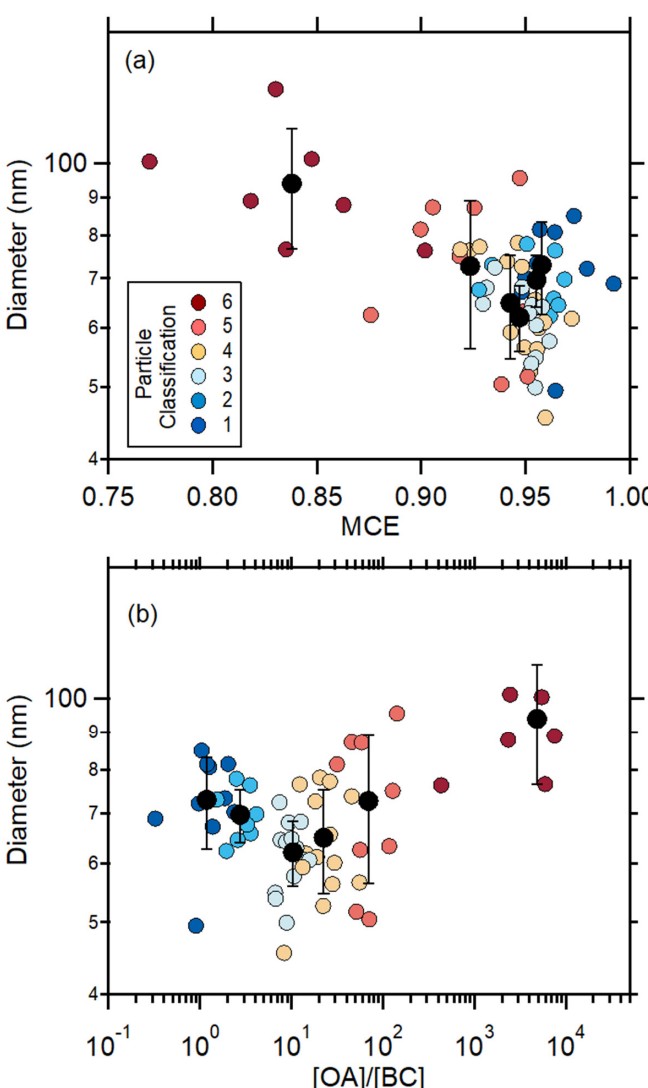


**Figure 9.** Relationship between number-weighted particle median diameter and (a) the MCE and (b) the [OA]/[BC] ratio. Colored circles are for individual burns and black circles for particle class averages.


**Figure 10.** Normalized total particle number-weighted (top) and the BC-only mass-weighted
(bottom) size distributions shown by fuel type (see legend). Individual burns are gray and averages
for a fuel type colors. For some fuels there is only one size distribution. Bimodal log-normal fits
are the black lines. The "other" category includes non-traditional biofuels, specifically building
materials and excelsior.
