# Peer review of "Biomass-burning derived particles from a wide variety of"

_Atmospheric Chemistry and Physics, 2019_

## Referee Comment (RC1) · Anonymous Referee #1 · 10 Sep 2019

General comments: The authors have comprehensively investigated the microphysical and optical properties of primary carbonaceous particles derived from various types of biomass-burning and their empirical relationships with some bulk parameters such as MCE and OA/BC ratio, on the basis of fire-chamber experiments. These results will be useful as a basis for interpreting the field-campaign data and for parameterization of size distribution and absorbing properties of OA, BrC, BC for biomass-burning plumes. The manuscript is logically written and display items are all easy to understand. However, I have a serious concern in the author's interpretation of their experimental results as detailed below. I can recommend publication of this manuscript after the authors convincingly address this issue.

[Figure]

Major critical comment: The authors observed the MAC_BC at 781 nm was nearly independent of Rcoat_BC. And they just mentioned that this negligible coating-induced absorption enhancement of BC was consistent with previous results by McMeeking et al. (2014), without providing detailed physical interpretations. To my intuition, the coating-induced enhancement for an absorbing core embedded inside a non-absorbing host particle is a general consequence of electromagnetics law (i.e., Maxwell equation), and should not be violated excepting very rare cases (I don't know any example of such cases). One possible condition potentially consistent with the negligible enhancement is that the measured rBC-containing aerosols are in the morphological form of "attached-type" rather than "coated-type". However, the attached-type assumption seems to be inconsistent with the principle of coating measurement using the SP-AMS, because the coating materials on rBC may not effectively vaporize in that type. The authors should provide convincing theoretical discussion supporting the author's assumption that the observed negligible coating-induced enhancement is a real physical phenomenon (and not a consequence of some measurement artifacts). In this paper, a convincing interpretation of the negligible coating-induced enhancement is also needed for supporting the robustness of the BrC estimate according to Eq.(6).

Minor comments Line 68. Typo: very -> vary
* * *

---

## Author Comment (AC1) · 10 Sep 2019

**Response to Reviewer #1**

We thank the reviewer for the thoughtful comments. The reviewer raises as a primary concern the measurements of the small absorption enhancement observed at 781 nm and the relationship with the coating-to-core ratio for BC-containing particles, positing that there may be some measurement bias that is leading to a strong deviation from the core-shell behavior.

The primary argument provided by the reviewer is that the SP-AMS might be biased because coatings may not effectively vaporize if they are not engulfing the BC. Certainly this is a possibility. However, we note that if this occurred it would lead to a negative bias and thus the reported coating-to-core ratios would be a lower bound. This would seem to go opposite to the reviewers concern; if the actual coating-to-core ratios were even larger than reported then the disparity only increases. It is true that there is a differential sensitivity of the SP-AMS to BC compared to coating materials owing to how the particle beam overlaps with the laser beam in the instrument (Willis et al., 2014). However, this is more important for absolute quantification than it is for relative quantification (i.e., coating-to-core ratios). Based on Willis et al. (2014), if no accounting of the coating dependence of the detection were accounted for, such detection issues could lead to a bias of ca. 30% in the coating-to-core ratios. While important, such a bias would not materially affect the conclusions here.

We also note that there is an abundance of evidence in the literature for non-core-shell morphologies for fresh biomass-derived BC-containing particles. A just published paper from Adachi et al. (2019) shows images of BC attached to other material, consistent with some of their previous work (Adachi and Buseck, 2008;Adachi et al., 2010;Adachi and Buseck, 2011) and with various other similar measurements (Chakrabarty et al., 2006;China et al., 2013;Torvela et al., 2014). Using an SP2, Sedlacek et al. (2012) observed evidence of non-core-shell morphologies for BC-containing particles in a biomass burning plume, with the fraction of such particles >60% even in a somewhat aged plume. Also with an SP2, (Pan et al., 2017) observed fresh biomass combustion-derived large BC, averaged over many different fuel types and for large (>200 nm diameter) BC cores, exhibits a wide range of estimated shell-to-core diameter ratios and "delay times" that correspond to coating-to-core mass ratios of ca. 0.1 to 3.

One might ask why BC-containing particles from biomass combustion would not readily adopt core-shell morphologies, as the above cited experimental evidence suggests? We suggest BC and coating material existing in the same particles most likely results from near-source coagulation. Sedlacek et al. (2015) observed formation of non-core-shell morphologies from coagulation, albeit not for biomass burning derived particles.

Additionally, it should be considered that the SP-AMS coating-to-core ratios reported here are bulk averages and do not account for the different mixing states of BC-containing particles. The distribution of coating material across the population of particles impacts the absorption enhancement, even when core-shell morphologies are assumed (Fierce et al., 2016). This was also shown in Cappa et al. (2012) for a simple 1:1 bimodal mixture of a mode having smaller coating-to-core ratios (1 or 0.1) and one having values that allowed for matching of the observed coating-to-core ratio. For mixtures of this sort, the predicted absorption enhancement is smaller than obtained if all particles are assumed equivalent. Thus, the issue is not simply one of morphology, but of particle-to-particle mixing state.

The reviewer asks for "convincing theoretical discussion." We lack sufficient information regarding the mixing state (i.e., the distribution of coatings with respect to the BC particle population) and internal

morphology to robustly calculate theoretical absorption enhancements for our experiments. However, example calculations following the approach of Cappa et al. (2012) can give an indication of what conditions might give rise to limited absorption enhancements, even at relatively large bulk-average coating-to-core ratios. Theoretical absorption enhancements, assuming core-shell morphologies, have been calculated for a binary population of particles, with one population "thinly" coated (with coating-to-core = 0.1) and one "thickly" coated (with variable coating-to-core ratios). The fraction of thickly coated particles ($f_{thick}$) was varied from 1 (for which the thickly coated coating-to-core ratio equals the bulk average) to 0.01. We have assumed a BC core diameter of 150 nm with a complex refractive index of $2.0 + 1.0i$. The complex RI for the coating was assumed as $1.5 + 10^{-8}i$. The absorption enhancement was calculated for each assumed $f_{thick}$ as a function of the bulk-average coating-to-core ratio. The results of these calculations are compared with the observations in Fig. 1a, and the variation in $E_{abs}$ with the $f_{thick}$ for three different bulk-average $R_{coat-BC}$ values (= 1, 5, and 10) are shown in Fig. 1b. These example calculations indicate that when the population is skewed towards most particles being thinly coated the theoretical $E_{abs}$ can be quite small even when the bulk-average $R_{coat-BC}$ is large. We note again that these calculations assume a core-shell morphology, so deviations from core-shell would serve to reduce these values further, that is the calculations here are upper-limits. Measurements by Liu et al. (2017) indicate that the core-shell approximation fails for particles having coating-to-core ratios < 3, and even above this value there may still be reductions owing to non-core-shell morphologies. Further, we have considered only a simple binary mixture of thinly and thickly coated particles. Consideration of more complex distributions of material across the BC population would lead to further reductions in the calculated absorption enhancements.

All this is to say that there is a strong experimental and theoretical foundation for observing absorption enhancements lower than the core-shell approximation. Given the simplistic nature of the calculations presented here, the assumptions that go into them, and a lack of experimental constraints regarding the particle mixing state for the particles sampled, we hesitate to add too much of this discussion to the manuscript. Nonetheless, in the revised version we intend to expand somewhat the discussion on Page 10 where we already indicated that the relatively low and constant $E_{abs}$ at 781 nm likely results from a combination of mixing state and morphology effects.

[Figure]

**Figure 1.** (a) The observed (points) absorption enhancement at 781 nm, calculated as the observed mass absorption coefficient divided by the reference value at the limit of no coating, and the calculated (lines) absorption enhancements from core-shell Mie theory as a function of the coating-to-core ratio. The different color lines correspond to different assumptions regarding the fraction of "thickly" coated BC. (b) The calculated absorption enhancement from core-shell Mie theory as a function of the fraction of "thickly" coated particles for three different bulk-average coating-to-core ratios.

**References**

Adachi, K., and Buseck, P. R.: Internally mixed soot, sulfates, and organic matter in aerosol particles from Mexico City, Atmospheric Chemistry and Physics, 8, 6469-6481, 2008.

Adachi, K., Chung, S. H., and Buseck, P. R.: Shapes of soot aerosol particles and implications for their effects on climate, J. Geophys. Res., 115, D15206, https://doi.org/10.1029/2009jd012868, 2010.

Adachi, K., and Buseck, P. R.: Atmospheric tar balls from biomass burning in Mexico, Journal of Geophysical Research-Atmospheres, 116, 7, https://doi.org/10.1029/2010jd015102, 2011.

Adachi, K., Sedlacek, A. J., Kleinman, L., Springston, S. R., Wang, J., Chand, D., Hubbe, J. M., Shilling, J. E., Onasch, T. B., Kinase, T., Sakata, K., Takahashi, Y., and Buseck, P. R.: Spherical tarball particles form

through rapid chemical and physical changes of organic matter in biomass-burning smoke, Proceedings of the National Academy of Sciences, 201900129, https://doi.org/10.1073/pnas.1900129116, 2019.

Cappa, C. D., Onasch, T. B., Massoli, P., Worsnop, D., Bates, T. S., Cross, E., Davidovits, P., Hakala, J., Hayden, K., Jobson, B. T., Kolesar, K. R., Lack, D. A., Lerner, B., Li, S. M., Mellon, D., Nuaanman, I., Olfert, J., Petaja, T., Quinn, P. K., Song, C., Subramanian, R., Williams, E. J., and Zaveri, R. A.: Radiative absorption enhancements due to the mixing state of atmospheric black carbon, Science, 337, 1078-1081, https://doi.org/10.1126/science.1223447, 2012.

Chakrabarty, R. K., Moosmüller, H., Garro, M. A., Arnott, W. P., Walker, J., Susott, R. A., Babbitt, R. E., Wold, C. E., Lincoln, E. N., and Hao, W. M.: Emissions from the laboratory combustion of wildland fuels: Particle morphology and size, Journal of Geophysical Research: Atmospheres, 111, D07204, https://doi.org/10.1029/2005jd006659, 2006.

China, S., Mazzoleni, C., Gorkowski, K., Aiken, A. C., and Dubey, M. K.: Morphology and mixing state of individual freshly emitted wildfire carbonaceous particles, Nat Commun, 4, https://doi.org/10.1038/ncomms3122, 2013.

Fierce, L., Bond, T. C., Bauer, S. E., Mena, F., and Riemer, N.: Black carbon absorption at the global scale is affected by particle-scale diversity in composition, Nat. Comm., 7, https://doi.org/10.1038/ncomms12361, 2016.

Liu, D. T., Whitehead, J., Alfarra, M. R., Reyes-Villegas, E., Spracklen, D. V., Reddington, C. L., Kong, S. F., Williams, P. I., Ting, Y. C., Haslett, S., Taylor, J. W., Flynn, M. J., Morgan, W. T., McFiggans, G., Coe, H., and Allan, J. D.: Black-carbon absorption enhancement in the atmosphere determined by particle mixing state, Nat. Geosci., 10, 184-U132, https://doi.org/10.1038/ngeo2901, 2017.

Pan, X., Kanaya, Y., Taketani, F., Miyakawa, T., Inomata, S., Komazaki, Y., Tanimoto, H., Wang, Z., Uno, I., and Wang, Z.: Emission characteristics of refractory black carbon aerosols from fresh biomass burning: a perspective from laboratory experiments, Atmos. Chem. Phys., 17, 13001-13016, https://doi.org/10.5194/acp-17-13001-2017, 2017.

Sedlacek, A. J., Lewis, E. R., Kleinman, L., Xu, J., and Zhang, Q.: Determination of and evidence for non-core-shell structure of particles containing black carbon using the Single-Particle Soot Photometer (SP2), Geophysical Research Letters, 39, n/a-n/a, https://doi.org/10.1029/2012GL050905, 2012.

Sedlacek, A. J., Lewis, E. R., Onasch, T. B., Lambe, A. T., and Davidovits, P.: Investigation of Refractory Black Carbon-Containing Particle Morphologies Using the Single-Particle Soot Photometer (SP2), Aerosol Science and Technology, 49, 872-885, https://doi.org/10.1080/02786826.2015.1074978, 2015.

Torvela, T., Tissari, J., Sippula, O., Kaivosoja, T., Leskinen, J., Virén, A., Lähde, A., and Jokiniemi, J.: Effect of wood combustion conditions on the morphology of freshly emitted fine particles, Atmospheric Environment, 87, 65-76, https://doi.org/10.1016/j.atmosenv.2014.01.028, 2014.

Willis, M. D., Lee, A. K. Y., Onasch, T. B., Fortner, E. C., Williams, L. R., Lambe, A. T., Worsnop, D. R., and Abbatt, J. P. D.: Collection efficiency of the soot-particle aerosol mass spectrometer (SP-AMS) for internally mixed particulate black carbon, Atmospheric Measurement Techniques, 7, 4507-4516, https://doi.org/10.5194/amt-7-4507-2014, 2014.

---

## Referee Comment (RC2) · Anonymous Referee #2 · 24 Sep 2019

The authors have investigated optical, physical, and chemical properties of primary aerosol particles generated from laboratory biomass burning of various fuels under different combustion conditions at the Missoula Fire Science Laboratory. The measured properties were correlated with bulk properties such as the MCE and OA/BC ratios. The authors concluded that intensive optical properties of aerosols are well parametrized using OA/BC mass ratios. The authors also observe negligible coating-induced black carbon absorption enhancements and little relationship between bulk OA chemical properties and OA/BC mass ratios. This is a dense manuscript; the results are presented in a coherent and logical manner and most figures are easy to follow. That said, I recommend major revision of this manuscript–my concerns and comments

are provided below.

Major comments:

1) First major concern is the source of black carbon (BC) constituting the Class 6 particles. Per table 1, the fuels burnt constitute duffs, peat, dung - all of which have been found to smolder (low MCE values) and produce tar-balls or spherical brown carbon (BrC) aerosol with negligible/no BC, very high SSA, and AAE >6 in the 405-532 nm. For example: Chakrabarty et al. ((2010), ACP 10, 6363) observed and reported no BC from duff burning at Missoula FSL. More recently, peat collected from Alaska and Indonesia were burned in a Missoula FSL-replica chamber (Sumlin et al. 2017 and 2018 series of papers) and negligible BC was found. These fuels have been only observed to smolder (low-temperature fires) both in the lab as well in field. Consequently, the particle formation mechanism is distinct in these fires, meaning soot (BC) formation is not supported.

2) Could charring of organics by the SP2 and/or SP-AMS be responsible for enhanced rBC concentration erroneously showing up in particle classes, especially in Class 6? This needs to be addressed. Sedlacek et al. (Aerosol Research Letters 52:15, 1345-1350) convincingly showed that initially near-IR transparent low-volatility compounds (fulvic and humic acid) particles at room temperature undergo chemical transformations as temperature is increased in a heated tube, creating new near-IR absorption transitions. They also say that this phenomenon enable SP2-induced charring of organic aerosol including tar-balls (akin to Class 6 particles in this article). Sedlacek et al. observed around 5-10% mass loading of rBC in case of fresh OA/tar balls resulting from SP2-induced charring through near-IR light absorption. The reviewer is suspicious that the authors erroneously report rBC concentration corresponding to Class 6 particles due to this phenomenon and then draw their conclusions. Please provide substantive proofs that no soot photometer induced artifact is involved during the experiments, especially for Class 6 particles. If no evidence can be provided, please remove Class 6 particles from all plots which have [OA]/[BC] as the x-axis.

3) The authors purport negligible absorption enhancement at 781 nm for Rcoat values as large as 10 based on results from Figure 4c. My concern with this assertion is that the axes in these graphs are extremely skewed which can misrepresent the actual MACBC enhancements. The average Eabs for Rcoat less than 10 is mentioned to be close to 1.2, but theoretical Eabs for longer wavelengths at these coating values are not expected to exceed 2 regardless (see Chakrabarty and Heinson, Phys. Rev. Lett., 2018). I believe that if the axes were not disproportionately skewed due to the extremely large MACBC values at the large OA/BC mass ratios (corresponding to Class 6 particles which in turn are due to the very small BC concentrations rather than large OA concentrations) we would be able to discern larger coating-induced absorption enhancements even at 781 nm. The conclusion that there is negligible coating-induced absorption enhancement based on visual comparison with a skewed axis is in my opinion is highly misleading. I notice that there are points in Fig 4c which have MACBC values larger than 10 which in turn would correspond to Eabs close to 2 which is significant in terms of absorption enhancements.

4) The authors claim that the increased MACBC at 781 nm is due to OA absorption and not coating-induced. They need to cite relevant literature which demonstrates significant BrC absorption at longer wavelengths to back up this assertion.

5) In Figure 4, it does not make sense to include points for MACBC where the contribution of BrC to total absorption is much larger than that of BC. So, removing all points for Class 6 OA would make the plots in Figure 4 more informative. The BC concentration in Class 6 is very likely an artifact. The absorption enhancement at longer wavelengths have a weaker dependence on coating thickness than at shorter wavelengths as observed by Pokhrel et al. (2017) cited in the manuscript, but it is still significant. I am unconvinced of the insignificance of coating-induced BC light absorption enhancement asserted by the manuscript or at least from the results as they have been presented right now.

6) Title should bear the word "laboratory" since the observed results might not be appli-

cable to real world fires. For e.g., something like: "Laboratory-based biomass burning particles from a wide variety of fuels: Part 1: Properties of primary particles"

7) The authors provide no explanation (beyond a hand waving argument) to back the statement "The contribution of coating-induced enhancements (i.e. lensing effects) to absorption by black carbon are shown to be negligible for all conditions". Lensing or focusing of light to the core could also be possible with weakly light-absorbing coating materials such as brown carbon with low imaginary index of refraction. Such a coating would facilitate lensing in addition to itself absorbing. One convincing way to declare that "no lensing" takes place is by looking at the internal field strength of a brown-carbon coated BC aggregate (see methodology in Chakrabarty and Heinson, Phys. Rev. Lett, 2018). I would like to see such a rigorous analysis performed (DDA or T-Matrix and not Mie-based core-shell) by the authors on a few BC aggregates coated with BrC vs non-refractory materials and convincing the reviewer and the community if indeed the "no lensing" claim is valid. If the authors cannot perform such an analysis, then I suggest that they remove all statements from the abstract and the main text regarding "negligible coating-induced enhancements (lensing effects)". Instead, rephrase or replace the sentences with "brown carbon-coated BC particles yield absorption enhancements of x and y values..."

Minor comments:

L68: change very to vary Add units for MACBC at all relevant plots. L164: When the authors say that the relationship between SSA and OA/BC is consistent with those of Pokhrel at al. could they expand on whether the corresponding fits parameters in the two studies match as well and if they do not then possible reasons for the mismatch. L169: change Table S1 to Table S2 L330: change Eqn. 7 to Eqn. 6

---

## Author Comment (AC2) · 1 Nov 2019

Response to Reviewer #2:

We thank the reviewer for the thoughtful comments regarding, among other issues, potential biases in the BC measurements and in the absorption measurements. Certainly such issues are important to consider. Below we provide a point-by-point response in which we argue that the potential issues raised by the reviewer did not impact our observations and that our interpretations are robust. Our responses are in **blue** and the initial reviewer comments in **black**.

1. First major concern is the source of black carbon (BC) constituting the Class 6 particles. Per table 1, the fuels burnt constitute duffs, peat, dung - all of which have been found to smolder (low MCE values) and produce tar-balls or spherical brown carbon (BrC) aerosol with negligible/no BC, very high SSA, and AAE >6 in the 405-532 nm. For example: Chakrabarty et al. ((2010), ACP 10, 6363) observed and reported no BC from duff burning at Missoula FSL. More recently, peat collected from Alaska and In- donesia were burned in a Missoula FSL-replica chamber (Sumlin et al. 2017 and 2018 series of papers) and negligible BC was found. These fuels have been only observed to smolder (low-temperature fires) both in the lab as well in field. Consequently, the particle formation mechanism is distinct in these fires, meaning soot (BC) formation is not supported.

Response: The reviewer here is arguing that there is zero BC produced. Later (Comments 4) the reviewer questions whether the OA can be absorbing at 781 nm and suggests that there is a more notable enhancement of BC absorption at 781 nm at very large Rcoat values. We find these arguments to be somewhat inconsistent. The very large Rbc values are determined for the systems where the total [OA]/[BC] ratios are largest. These correspond to the much more smoldering burns. If there is no BC (as the reviewer suggests) then there can be no enhancement of BC absorption and, related, if the OA is not absorbing then there should be no absorption at all if there is no BC. Yet, we clearly observe absorption at 781 nm for the high [OA]/[BC] systems (i.e., Class 6). Thus, we must conclude that, at minimum, there is either some small amount of BC for the Class 6 particles or the OA is somewhat absorbing at 781 nm. We believe our results suggest both to be true.

First, where the reviewer writes that Chakrabarty et al. (2010) "observed and reported no BC from duff burning," we note that the cited paper states only that "A statistically relevant number of particles have been examined for morphology using SEM, and it was found that a high fraction (>95%) of all particles from each of the three samples were tar balls." It is not stated what the other 5% of particles corresponded to. It is stated that thermal EC was measured for these same samples but "below detection limit." However, this is, perhaps, not unexpected given that the total OA/BC mass ratios we derived were larger than 1000 for these very OA-rich particles. The amount of BC (or EC) is indeed, quite small, and depending on the detection limit of the OC/EC instrument used any EC present might not be quantified. Further, one would need fewer than 1 in 1000, or even 1 in 10,000, particles to be BC (assuming the same mass-per-particle) for our results to hold. The total number of particles analyzed by Chakrabarty et al. (2010) was not reported, but we believe it quite reasonable to think that 1 in 1,000 particles could have been BC-containing. Additionally, we note that the analysis of Chakrabarty et al. (2010) shown in Fig. 4 and Eqn. 3 seems to implicitly assume that there is a BC contribution, which they note resulted from "minor flaming combustion during the ignition of the fire." Looking additionally to

Chakrabarty et al. (2016), who characterized emissions from Alaskan and Siberian peat, their Table 1 explicitly shows that BC is emitted. They report BC emission factors of 0.1-0.2 g kg$^{-1}$ fuel, compared to OC emission factors of 4-7 g kg$^{-1}$ fuel, corresponding to OA-to-BC ratios (assuming an OA/OC ratio of 1.6) of 32-176, which are smaller even than what we report. (We note that the actual results reported in Chakrabarty et al. (2016) contrasts with what is stated in Sumlin et al. (2017) where Sumlin (2017) state that for Chakrabarty (2016) "smoldering Alaskan and Siberian peat emissions contain BrC aerosols with no BC component.") There are also the results of Bhattarai et al. (2018), who characterized smoke from combustion of three different peats. They used EC/OC analysis for EC concentrations, and a PASS-3 for absorption. The EF's for EC are small (0.01-0.1 mg/g fuel), with OA/EC ratios of 154-522 (again assuming OA/OC = 1.6). Again, the amount of EC emitted is small, but not zero. For comparison, the [OA]/[BC] ratios we determined are even larger than this.  The reviewer notes that Sumlin et al. (2017,2018) found "negligible" BC; it is not clear to us where this conclusion arises from since the reported measurement suite in Sumlin et al. (2017, 2018) did not include instrumentation for measurement of BC as best we can tell. Regardless, we are not arguing that there is a lot of BC here, and with an [OA]/[BC] > 1000 some might consider the amount of BC negligible. But, a "negligible amount" does not imply that BC is non-existent, and indeed there are literature results (e.g., Chakrabarty et al. (2016) and Bhattarai et al. (2018)) supporting the idea that there is a small amount of BC emitted from peat combustion.

Second, as to whether the OA is absorbing at 781 nm, we believe our observations are clear. If, as the reviewer contends, there is no BC present for these very OA-rich particles (which we do not think to be the case; see above) then there should be no absorption at 781 nm if they are not absorbing. Yet, the observed absorption at 781 nm was well above the detection limit. Thus, if there is no BC present the OA must be absorbing. (Also, if the OA is not absorbing at 781 nm, then the absorption at 1064 nm might be similarly small, and thus charring would not be expected.) But, as we argue, there is some small amount of BC present. Unless we are dramatically underestimating the amount of BC present in these very OA-rich particles then the magnitude of the derived MAC_BC values for these particular particles are too large (>100 m2/g in one case) to be reasonably explained through coating effects. Thus, again, the OA must be absorbing.

2. Could charring of organics by the SP2 and/or SP-AMS be responsible for enhanced rBC concentration erroneously showing up in particle classes, especially in Class 6? This needs to be addressed. Sedlacek et al. (Aerosol Research Letters 52:15, 1345- 1350) convincingly showed that initially near-IR transparent low-volatility compounds (fulvic and humic acid) particles at room temperature undergo chemical transforma- tions as temperature is increased in a heated tube, creating new near-IR absorption transitions. They also say that this phenomenon enable SP2-induced charring of or- ganic aerosol including tarballs (akin to Class 6 particles in this article). Sedlacek et al. observed around 5-10% mass loading of rBC in case of fresh OA/tar balls resulting from SP2-induced charring through near-IR light absorption. The reviewer is suspi- cious that the authors erroneously report rBC concentration corresponding to Class 6 particles due to this phenomenon and then draw their conclusions. Please provide substantive proofs that no soot photometer induced artifact is involved during the experiments, especially for Class 6 particles. If no evidence can be provided, please remove Class 6 particles from all plots which have [OA]/[BC] as the x-axis.

Response: The work of Sedlacek et al. (2018) is indeed important to consider. First, we believe it is very important to recognize that Sedlacek et al. (2018) only report that the SP2 detects fulvic acid and humic acid, BrC surrogates, as rBC when they are heated in a tube furnace to >500 degrees C. Their Fig. 2 shows that as this pre-heating temperature is reduced the likelihood of rBC detection is reduced. Indeed, it is implied in their paper (although not explicitly stated) that without heating neither fulvic acid or humic acid are detected as rBC in the SP2. In our experiments, the particles were not heated prior to detection with the SP2 (excluding the heating inherent in the particle generation). Thus, if our particles behave as fulvic or humic acid then we would not expect charring to be a concern.

Sedlacek et al. (2018) did also investigate charring of quite absorbing lab-generated tar balls, which is potentially of more relevance to our experiments than fulvic or humic acid samples. However, as they note "recent field observations suggest ambient tar balls may be less absorbing ($k \sim 0.02i$ at 532 nm) than laboratory tar balls," with the latter having values around $0.2i$. Our median derived $MAC_{OA}$ at 532 nm was 0.21 m$^2$/g, corresponding to an imaginary RI of around $0.007i$. Thus, our particles are more similar to field tar balls than lab-generated tar balls. Importantly, the tarballs investigated in Sedlacek et al. (2018) are not "akin to the Class 6 particles" as suggested by the reviewer. The reason for this is almost certainly the completely different particle production methods used in in our study versus by Sedlacek et al. (2018). Given the very different production methods, it is to be expected that the particle chemical properties differ, especially the graphitic content (which is likely of importance to any bias in the SP2 analysis).

3. The authors purport negligible absorption enhancement at 781 nm for Rcoat values as large as 10 based on results from Figure 4c. My concern with this assertion is that the axes in these graphs are extremely skewed which can misrepresent the actual MACBC enhancements. The average Eabs for Rcoat less than 10 is mentioned to be close to 1.2, but theoretical Eabs for longer wavelengths at these coating values are not expected to exceed 2 regardless (see Chakrabarty and Heinson, Phys. Rev. Lett., 2018). I believe that if the axes were not disproportionately skewed due to the extremely large MACBC values at the large OA/BC mass ratios (corresponding to Class 6 particles which in turn are due to the very small BC concentrations rather than large OA concentrations) we would be able to discern larger coating-induced absorption enhancements even at 781 nm. The conclusion that there is negligible coating-induced absorption enhancement based on visual comparison with a skewed axis is in my opinion is highly misleading. I notice that there are points in Fig 4c which have MACBC values larger than 10 which in turn would correspond to Eabs close to 2 which is significant in terms of absorption enhancements.

Recognizing the challenge of viewing things on multiple scales, we also included a version of this figure as Fig. S1, where results for each wavelength are shown on their own scale. We believe that there is also value in showing the results at the three different wavelengths on a common scale to visually illustrate the different behavior, and thus provided these two ways of viewing things. To the reviewers contention that we came to our conclusions based on "visual comparison with a skewed axis," this is simply not true. We came to our conclusion based on explicit calculation of the Eabs values from the observed $MAC_{BC}$ values and the extrapolated value at zero coating and interrogation of these calculated values

compared to the value determined from extrapolation to zero OA. As the reviewer notes, indeed there are $MAC_{BC}$ values much larger than 10 in Fig. 4c at 781 nm. The reviewer implies that this results from a significant absorption enhancement. However, our observations are much more consistent with these large $MAC_{BC}$ values resulting from OA absorption. As discussed above, even very weak absorption matters when the total [OA]/[BC] is large. This is why there is a much stronger relationship between the $MAC_{BC}$ and the total [OA]/[BC] than there is with the [coating]/[BC] ratio.  Further, we note that the largest $MAC_{BC}$ values correspond to $E_{abs}$ values > 10 (from $MAC_{BC}$ values > 100 $m^2$/g). As the reviewer agrees, $E_{abs}$ values from non-absorbing coatings on BC "are not expected to exceed 2". Thus, we must conclude that the observable enhancement includes an important contribution from BrC absorption. Finally, we note that if, as the reviewer contends above (although we disagree with), the [BC] are overestimated for Class 6 particles then the reported $MAC_{BC}$ are underestimated for this class of particles, implying an even larger contribution from BrC. In any case, we have updated the manuscript to be more quantitative regarding the observed $E_{abs}$ at 781 nm, adding the new text provided in our response to point 6 below.

4. The authors claim that the increased MACBC at 781 nm is due to OA absorption and not coating-induced. They need to cite relevant literature which demonstrates significant BrC absorption at longer wavelengths to back up this assertion.

While we believe our observations are clear on their own (see above discussion), we are happy to cite relevant literature. If the reviewer has any particular studies in mind, we would be happy to include them. Otherwise, we can include (for example) the classic paper of Kirchstetter et al. (2004), who report absorption by OC out to at least 700 nm, along with some others (Alexander et al., 2008;Phillips and Smith, 2017;Sengupta et al., 2018;Sumlin et al., 2018). We have added these to Table S3.

We also make the simple argument here: various studies indicate that absorption by BrC declines reasonably continuously with increasing wavelength. So, a thought experiment. If the $MAC_{OA405nm}$ = 1 $m^2$ $g^{-1}$ and the AAE = 5, simple extrapolation (assuming a constant AAE) yields an $MAC_{OA,781nm}$ = 0.037 $m^2$ $g^{-1}$. This is small, but not zero and, when the OA concentration is much larger than the BC concentration, should be readily observable. If we instead assume the larger AAE values observed in our study, ~8.5, the extrapolated $MAC_{OA,781nm}$ = 0.01 $m^2$ $g^{-1}$. A key point is that when the OA abundance is sufficiently large even weak OA absorption can matter. This can be looked at one additional way. Consider that the absorption ratio between BC and OA is equal to ([BC]*$MAC_{BC}$/[OA]*$MAC_{BrC}$). If the [BC]/[OA] ratio is $10^{-3}$ (which we observe) then the OA absorption will be observable even if the ratio $MAC_{BC}$/$MAC_{BrC}$ is 1000. Given an $MAC_{BC}$ ~4 $m^2$/g at 781 nm, this means the $MAC_{BrC}$ need only be 0.004 $m^2$/g to matter at the largest [OA]/[BC]. We contend it is quite reasonable to think that some BrC is at least this absorbing at 781 nm. Such exceptionally small absorption might be true for some secondary OA, it seems less likely for OA from biomass combustion, which typically has larger $MAC_{OA}$ values compared to SOA (see Lambe et al. (2013)).

5. In Figure 4, it does not make sense to include points for MACBC where the contribution of BrC to total absorption is much larger than that of BC. So, removing all points for Class 6 OA would make the plots in Figure 4 more informative.  The BC concentration  in Class 6 is very likely an artifact. The absorption enhancement at longer wavelengths have a weaker dependence on coating thickness than at shorter wavelengths as ob- served by Pokhrel et

al. (2017) cited in the manuscript, but it is still significant. I am unconvinced of the insignificance of coating-induced BC light absorption enhancement asserted by the manuscript or at least from the results as they have been presented right now.

First, as we discuss extensively above, the BC concentration is Class 6 is not likely an artifact, and thus removal of these points is not warranted. Second, we wish to clarify that nowhere do we conclude that the absorption enhancement is "insignificant." We did, however, state it is "negligible," and we believe this consistent with our observations. Perhaps this is parsing words, but we believe there is a difference between "negligible" and "insignificant." The median $E_{abs}$ (based on the ratio of $MAC_{BC}$ values), excluding the values that are exceptionally large (>3, and almost certainly dominated by OA absorption) was 1.14 and the mean was 1.17. These are "significantly" greater than one (in the statistical sense), yet still, in our view, "negligible." However, we have revised the language in the paper to note that there is "only a minor coating-induced enhancement," rather than a "negligible" enhancement and to make our statements more quantitative. Some of this was already discussed in Section 3.4.2, where we reported the mean $E_{abs}$ at 781 nm. However, we have added an additional paragraph at the end of Section 3.4.1.

> "Values for the absorption enhancement at 781 nm are calculated as the ratio between the observed $MAC_{BC}$ in **Figure 4** and the derived $MAC_{BC,pure}$. The derived $E_{abs}$ range from 0.96 to 27. Values greater than two occur only for the particles having particularly large [OA]/[BC], > 400. As $E_{abs}$ values much greater than two at 781 nm are unlikely to result from mixing-induced enhancements, this again suggests that the OA is somewhat absorbing at this wavelength. For the burns where [OA]/[BC] < 400, the median $E_{abs}$ = 1.14 and the arithmetic mean $E_{abs}$ = 1.19 ± 0.14 (1σ). Given that some of this enhancement may result from BrC absorption at 781, these values can be considered upper-limits on $E_{abs,coat}$, and the small magnitude is consistent with our conclusion above that, while likely greater than zero, the mixing-induced enhancement is generally negligible. It is possible that the $E_{abs,coat}$ values when [OA]/[BC] > 400 are substantially larger. However, given the general lack of a dependence of the $MAC_{BC,781nm}$ for $R_{BC-coat}$ < 10 this seems unlikely."

Alexander, D. T. L., Crozier, P. A., and Anderson, J. R.: Brown Carbon Spheres in East Asian Outflow and Their Optical Properties, Science, 321, 833-836, https://doi.org/10.1126/science.1155296, 2008.
Bhattarai, C., Samburova, V., Sengupta, D., Iaukea-Lum, M., Watts, A. C., Moosmuller, H., and Khlystov, A. Y.: Physical and chemical characterization of aerosol in fresh and aged emissions from open combustion of biomass fuels, Aerosol Science and Technology, 52, 1266-1282, https://doi.org/10.1080/02786826.2018.1498585, 2018.
Chakrabarty, R. K., Moosmüller, H., Chen, L. W. A., Lewis, K., Arnott, W. P., Mazzoleni, C., Dubey, M. K., Wold, C. E., Hao, W. M., and Kreidenweis, S. M.: Brown carbon in tar balls from smoldering biomass combustion, Atmospheric Chemistry and Physics, 10, 6363-6370, https://doi.org/10.5194/acp-10-6363-2010, 2010.
Chakrabarty, R. K., Gyawali, M., Yatavelli, R. L. N., Pandey, A., Watts, A. C., Knue, J., Chen, L. W. A., Pattison, R. R., Tsibart, A., Samburova, V., and Moosmuller, H.: Brown carbon aerosols from burning of boreal peatlands: microphysical properties, emission factors, and implications for direct radiative forcing, Atmospheric Chemistry and Physics, 16, 3033-3040, https://doi.org/10.5194/acp-16-3033-2016, 2016.

Kirchstetter, T. W., Novakov, T., and Hobbs, P. V.: Evidence that the spectral dependence of light absorption by aerosols is affected by organic carbon, Journal of Geophysical Research-Atmospheres, 109, D21208, 2004.

Lambe, A. T., Cappa, C. D., Massoli, P., Onasch, T. B., Forestieri, S. D., Martin, A. T., Cummings, M. J., Croasdale, D. R., Brune, W. H., Worsnop, D. R., and Davidovits, P.: Relationship between Oxidation Level and Optical Properties of Secondary Organic Aerosol, Environmental Science & Technology, 47, 6349-6357, https://doi.org/10.1021/es401043j, 2013.

Phillips, S. M., and Smith, G. D.: Spectroscopic comparison of water- and methanol-soluble brown carbon particulate matter, Aerosol Science and Technology, 51, 1113-1121, https://doi.org/10.1080/02786826.2017.1334109, 2017.

Sedlacek, A. J., Onasch, T. B., Nichman, L., Lewis, E. R., Davidovits, P., Freedman, A., and Williams, L.: Formation of refractory black carbon by SP2-induced charring of organic aerosol, Aerosol Science and Technology, 52, 1345-1350, https://doi.org/10.1080/02786826.2018.1531107, 2018.

Sengupta, D., Samburova, V., Bhattarai, C., Kirillova, E., Mazzoleni, L., Iaukea-Lum, M., Watts, A., Moosmüller, H., and Khlystov, A.: Light absorption by polar and non-polar aerosol compounds from laboratory biomass combustion, Atmos. Chem. Phys., 18, 10849-10867, https://doi.org/10.5194/acp-18-10849-2018, 2018.

Sumlin, B. J., Pandey, A., Walker, M. J., Pattison, R. S., Williams, B. J., and Chakrabarty, R. K.: Atmospheric Photooxidation Diminishes Light Absorption by Primary Brown Carbon Aerosol from Biomass Burning, Environmental Science & Technology Letters, 4, 540-545, https://doi.org/10.1021/acs.estlett.7b00393, 2017.

Sumlin, B. J., Heinson, Y. W., Shetty, N., Pandey, A., Pattison, R. S., Baker, S., Hao, W. M., and Chakrabarty, R. K.: UV–Vis–IR spectral complex refractive indices and optical properties of brown carbon aerosol from biomass burning, Journal of Quantitative Spectroscopy and Radiative Transfer, 206, 392-398, https://doi.org/10.1016/j.jqsrt.2017.12.009, 2018.

---

## Author Response (AR1)

**UNIVERSITY OF CALIFORNIA, DAVIS**

[Figure]

COLLEGE OF ENGINEERING

DEPARTMENT OF CIVIL & ENVIRONMENTAL ENGINEERING
ONE SHIELDS AVENUE
DAVIS, CALIFORNIA 95616
PHONE (530) 752-8180
FAX (530) 752-7872

November 2019

Dear Dr. Shiraiwa,

Below please find our point-by-point responses to the reviewers, along with a tracked changes
version of our manuscript. The reviewers raised important points regarding data quality and
potential measurement biases. We have considered these issues thoroughly and concluded that
our measurements are robust . We have modified our manuscript accordingly.

Should you require further information or more detailed responses to help you make your
decision we would be happy to provide.

Best regards,

Christopher D. Cappa
Ray B. Krone Professor of Environmental Engineering

**Response to Reviewer #1**

We thank the reviewer for the thoughtful comments. The reviewer raises as a primary concern the measurements of the small absorption enhancement observed at 781 nm and the relationship with the coating-to-core ratio for BC-containing particles, positing that there may be some measurement bias that is leading to a strong deviation from the core-shell behavior. Our responses are in **blue** and the initial reviewer comments in **black**.

General comments: The authors have comprehensively investigated the microphysical and optical properties of primary carbonaceous particles derived from various types of biomass-burning and their empirical relationships with some bulk parameters such as MCE and OA/BC ratio, on the basis of fire-chamber experiments. These results will be useful as a basis for interpreting the field-campaign data and for parameterization of size distribution and absorbing properties of OA, BrC, BC for biomass-burning plumes. The manuscript is logically written and display items are all easy to understand. How- ever, I have a serious concern in the author's interpretation of their experimental results as detailed below. I can recommend publication of this manuscript after the authors convincingly address this issue.

The primary argument provided by the reviewer is that the SP-AMS might be biased because coatings may not effectively vaporize if they are not engulfing the BC. Certainly this is a possibility. However, we note that if this occurred it would lead to a negative bias and thus the reported coating-to-core ratios would be a lower bound. This would seem to go opposite to the reviewers concern; if the actual coating-to-core ratios were even larger than reported then the disparity only increases. It is true that there is a differential sensitivity of the SP-AMS to BC compared to coating materials owing to how the particle beam overlaps with the laser beam in the instrument (Willis et al., 2014). However, this is more important for absolute quantification than it is for relative quantification (i.e., coating-to-core ratios). Based on Willis et al. (2014), if no accounting of the coating dependence of the detection were accounted for, such detection issues could lead to a bias of ca. 30% in the coating-to-core ratios. While important, such a bias would not materially affect the conclusions here.

We also note that there is an abundance of evidence in the literature for non-core-shell morphologies for fresh biomass-derived BC-containing particles. A just published paper from Adachi et al. (2019) shows images of BC attached to other material, consistent with some of their previous work (Adachi and Buseck, 2008;Adachi et al., 2010;Adachi and Buseck, 2011) and with various other similar measurements (Chakrabarty et al., 2006;China et al., 2013;Torvela et al., 2014). Using an SP2, Sedlacek et al. (2012) observed evidence of non-core-shell morphologies for BC-containing particles in a biomass burning plume, with the fraction of such particles >60% even in a somewhat aged plume. Also with an SP2, (Pan et al., 2017) observed fresh biomass combustion-derived large BC, averaged over many different fuel types and for large (>200 nm diameter) BC cores, exhibits a wide range of estimated shell-to-core diameter ratios and "delay times" that correspond to coating-to-core mass ratios of ca. 0.1 to 3.

One might ask why BC-containing particles from biomass combustion would not readily adopt core-shell morphologies, as the above cited experimental evidence suggests? We suggest BC and coating material existing in the same particles most likely results from near-source coagulation. Sedlacek et al. (2015) observed formation of non-core-shell morphologies from coagulation, albeit not for biomass burning derived particles.

Additionally, it should be considered that the SP-AMS coating-to-core ratios reported here are bulk averages and do not account for the different mixing states of BC-containing particles. The distribution of coating material across the population of particles impacts the absorption enhancement, even when core-shell morphologies are assumed (Fierce et al., 2016). This was also shown in Cappa et al. (2012) for a simple 1:1 bimodal mixture of a mode having smaller coating-to-core ratios (1 or 0.1) and one having values that allowed for matching of the observed coating-to-core ratio. For mixtures of this sort, the predicted absorption enhancement is smaller than obtained if all particles are assumed equivalent. Thus, the issue is not simply one of morphology, but of particle-to-particle mixing state.

Major critical comment: The authors observed the MAC_BC at 781 nm was nearly in- dependent of Rcoat_BC. And they just mentioned that this negligible coating-induced absorption enhancement of BC was consistent with previous results by McMeeking et al. (2014), without providing detailed physical interpretations. To my intuition, the coating-induced enhancement for an absorbing core embedded inside a non-absorbing host particle is a general consequence of electromagnetics law (i.e., Maxwell equation), and should not be violated excepting very rare cases (I don't know any example of such cases). One possible condition potentially consistent with the negligible enhancement is that the measured rBC-containing aerosols are in the morphological form of "attached-type" rather than "coated-type". However, the attached-type assumption seems to be inconsistent with the principle of coating measurement using the SP-AMS, because the coating materials on rBC may not effectively vaporize in that type. The authors should provide convincing theoretical discussion supporting the author's as- sumption that the observed negligible coating-induced enhancement is a real physical phenomenon (and not a consequence of some measurement artifacts). In this paper, a convincing interpretation of the negligible coating-induced enhancement is also needed for supporting the robustness of the BrC estimate according to Eq.(6).

The reviewer asks for "convincing theoretical discussion." We lack sufficient information regarding the mixing state (i.e., the distribution of coatings with respect to the BC particle population) and internal morphology to robustly calculate theoretical absorption enhancements for our experiments. However, example calculations following the approach of Cappa et al. (2012) can give an indication of what conditions might give rise to limited absorption enhancements, even at relatively large bulk-average coating-to-core ratios. Theoretical absorption enhancements, assuming core-shell morphologies, have been calculated for a binary population of particles, with one population "thinly" coated (with coating-to-core = 0.1) and one "thickly" coated (with variable coating-to-core ratios). The fraction of thickly coated particles ($f_{thick}$) was varied from 1 (for which the thickly coated coating-to-core ratio equals the bulk average) to 0.01. We have assumed a BC core diameter of 150 nm with a complex refractive index of $2.0 + 1.0i$. The complex RI for the coating was assumed as $1.5 + 10^{-8}i$. The absorption enhancement was calculated for each assumed $f_{thick}$ as a function of the bulk-average coating-to-core ratio. The results of these calculations are compared with the observations in Fig. 1a, and the variation in $E_{abs}$ with the $f_{thick}$ for three different bulk-average $R_{coat-BC}$ values (= 1, 5, and 10) are shown in Fig. 1b. These example calculations indicate that when the population is skewed towards most particles being thinly coated the theoretical $E_{abs}$ can be quite small even when the bulk-average $R_{coat-BC}$ is large. We note again that these calculations assume a core-shell morphology, so deviations from core-shell would serve to reduce these values further, that is the calculations here are upper-limits. Measurements by Liu et al. (2017) indicate that the core-shell approximation fails for particles having coating-to-core ratios < 3, and even above this value there may still be reductions owing to non-core-shell morphologies. Further, we have considered only a simple binary mixture of thinly and thickly coated particles. Consideration of more complex distributions of material across the BC population would lead to further reductions in the calculated absorption enhancements.

All this is to say that there is a strong experimental and theoretical foundation for observing absorption enhancements lower than the core-shell approximation. Given the simplistic nature of the calculations presented here, the assumptions that go into them, and a lack of experimental constraints regarding the particle mixing state for the particles sampled, we hesitate to add too much of this discussion to the manuscript. Nonetheless, in the revised version we intend to expand somewhat the discussion on Page 10 where we already indicated that the relatively low and constant $E_{abs}$ at 781 nm likely results from a combination of mixing state and morphology effects.

The reviewer also raises the question of whether the coating material vaporizes efficiently in the SP-AMS laser if the particles have an "attached-type" internal morphology. We, unfortunately, do not have evidence indicating whether such particles are or are not accurately characterized. However, we can consider what the impact would be on our results if this were a major issue. The reviewers concern seems to be that the coating material would be missed in the SP-AMS, although the rBC material should be detected. If this occurred, the bulk-average coating-to-core ratio observed would be underestimated. If this bias occurs (and again, we do not have evidence to argue one way or another whether it is a concern) then our coating-to-core estimates would be biased low, making the apparent gap between the core-shell expectation and the observed behavior even greater. Overall we agree with the reviewers comment that the presence of attached-type particles could contribute to the lower than expected enhancements, and imply as such when we stated that "Most likely, this lack of a substantial coating-induced enhancement results from a non-even distribution of non-BC mass across the population of BC particles (Fierce et al., 2016;Liu et al., 2017) and from the morphology of BC-containing particles not conforming to an idealized core-shell structure (Adachi et al., 2010).,

[Figure]

**Figure 1.** (a) The observed (points) absorption enhancement at 781 nm, calculated as the observed mass absorption coefficient divided by the reference value at the limit of no coating, and the calculated (lines) absorption enhancements from core-shell Mie theory as a function of the coating-to-core ratio. The different color lines correspond to different assumptions regarding the fraction of "thickly" coated BC. (b) The calculated absorption enhancement from core-shell Mie theory as a function of the fraction of "thickly" coated particles for three different bulk-average coating-to-core ratios.

[revised manuscript text omitted]

Formatted Table [1]

| | | | | | | |
|---|---|---|---|---|---|---|
| 405 | 0.037 | | Water soluble organic carbon | Regional | Kanpur, India | (Shamjad e... |
| 405 | | 0.7-1.3 | Water soluble organic carbon | Bonfire festival | Rehovot, Israel | (Bluvshtein... |
| 405 | | 0.6 | Methanol soluble organic carbon | Prescribed burn | NW US | (Xie et a... |
| 400, 600, 800 | 0.31, 0.26, 0.22 | | Electron loss | Asian outflow | Downwind of Asia | (Alexander... |
| 800/400 ratio | | 0.26 | Methanol soluble organic carbon | Ambient particles (ratio between wavelengths reported) | Athens, Georgia | (Phillips and... |
| 400, 550, 700 | 0.112, 0.030, 0.001 | | Acetone treatment + attenuation | African biomass burning | Southern Africa | (Kirchstetter... |

| Page 14: [1] Formatted Table | CDC | 10/29/2019 10:14:00 AM |
| --- | --- | --- |

Formatted Table

| Page 14: [2] Deleted | CDC | 10/29/2019 10:12:00 AM |
| --- | --- | --- |

$0.07 \pm 0.03/$
$0.04 \pm 0.01$

| Page 14: [3] Deleted | CDC | 10/29/2019 10:12:00 AM |
| --- | --- | --- |

Open fire/
Smoldering phase

---

## Referee Report (RR1)

In their rebuttal, McClure et al. didn't address points 6 and 7 of my first round comments (did I miss something here?). I consider both points to me major and important and hence, I ask them to address them again in this second round of revisions:

1) Title should bear the word "laboratory" since the observed results might not be applicable to real world fires. For e.g., something like: "Laboratory-based biomass burning particles from a wide variety of fuels: Part 1: Properties of primary particles"

2) The authors provide no explanation (beyond a hand waving argument) to back the statement "The contribution of coating-induced enhancements (i.e. lensing effects) to absorption by black carbon are shown to be negligible for all conditions". Lensing or focusing of light to the core could also be possible with weakly light-absorbing coating materials such as brown carbon with low imaginary index of refraction. Such a coating would facilitate lensing in addition to itself absorbing. The only convincing way to declare that "no lensing" takes place is by looking at the internal field strength of a brown-carbon coated BC aggregate (see methodology in Chakrabarty and Heinson, Phys. Rev. Lett, 2018). I would like to see such a rigorous analysis performed (DDA or T-Matrix and not Mie-based core-shell) by the authors on a few BC aggregates coated with BrC vs non-refractory materials and convincing the reviewer and the community if indeed the "no lensing" claim is valid. If the authors cannot perform such an analysis, then I suggest that they remove all statements from the abstract and the main text regarding "negligible coating-induced enhancements (lensing effects)". Instead, rephrase or replace the sentences with "brown carbon-coated BC particles yield absorption enhancements of x and y values…"

---

## Author Response (AR2)

Response to Reviewer #2:

Our apologies for missing comments 6 and 7 in our original response. (In converting the pdf to a word file we could copy from these were unintentionally lost by us.) We thank the reviewer for considering the revisions we have already made. Below we provide a response for both comments 6 and 7. Our responses are in **blue** and the initial reviewer comments in **black**.

1. Title should bear the word "laboratory" since the observed results might not be applicable to real world fires. For e.g., something like: "Laboratory-based biomass burning particles from a wide variety of fuels: Part 1: Properties of primary particles"

We understand the reviewers point here. However, we argue that there is nothing in the current title to indicate that these are "real-world" particles. (We assume by "real-world" the reviewer specifically means "ambient." The fires we sampled from were most assuredly "real.") In noting that the measurements are for "a wide variety of fuels" we believe it is implied that this is a laboratory experiment. Similarly, this is the case when we note the focus on "primary particles" as ambient studies have had limited success at accessing truly "primary" particles given the logistical challenges of getting close enough to fires and the rapid aging that occurs in the atmosphere. Instead of adjusting the title, we have updated the first sentence of the abstract to include the word "laboratory."

2. The authors provide no explanation (beyond a hand waving argument) to back the statement "The contribution of coating-induced enhancements (i.e. lensing effects) to absorption by black carbon are shown to be negligible for all conditions". Lensing or focusing of light to the core could also be possible with weakly light-absorbing coat- ing materials such as brown carbon with low imaginary index of refraction. Such a coating would facilitate lensing in addition to itself absorbing. One convincing way to declare that "no lensing" takes place is by looking at the internal field strength of a brown-carbon coated BC aggregate (see methodology in Chakrabarty and Heinson, Phys. Rev. Lett, 2018). I would like to see such a rigorous analysis performed (DDA or T-Matrix and not Mie-based core-shell) by the authors on a few BC aggregates coated with BrC vs non-refractory materials and convincing the reviewer and the community if indeed the "no lensing" claim is valid. If the authors cannot perform such an anal- ysis, then I suggest that they remove all statements from the abstract and the main text regarding "negligible coating-induced enhancements (lensing effects)". Instead, rephrase or replace the sentences with "brown carbon-coated BC particles yield ab- sorption enhancements of x and y values. . ."

This particular comment follows directly from the Reviewer's comment #3, questioning whether our conclusion that the "lensing" enhancement is "negligible." However, here the reviewer (i) asks for calculations including BrC and (ii) indicates that if we cannot do such calculations we should entirely revise the discussion of the absorption enhancement. We address the question of whether the lensing-induced absorption enhancement is "negligible" extensively in our original response to the Reviewer's comment #3. While we concluded that use of the "negligible" is justified, we have nonetheless revised the manuscript to indicate it as "minor". (We have additionally gone through now and done a search for "negligible," replacing it with "minor.") Here, we address more specifically the utility or appropriateness of inclusion of calculations such as those suggested. In short, we do not think that including results of such calculations would add substantial value to the manuscript. One reason is that we do not have knowledge of the internal morphology of the BC-containing particles. Given this absence of knowledge, it would be necessary for us to perform systematic calculations over different particle geometries and coating amounts to glean insights. However, such calculations have already been performed by many researchers and results are available in the literature, both for non-absorbing and weakly absorbing coatings (Adachi et al., 2010;Kahnert and Devasthale, 2011;Kahnert et al., 2012;Liu et al., 2015;Kahnert, 2017;Liu et al., 2017a;Zhang et al., 2017;Kanngiesser and Kahnert, 2018;Zhang et al., 2018). Performing new calculations here is therefore, in our opinion, unnecessary. Instead, we believe it sufficient to point to the literature, which we have done by citing the relatively early work by Adachi et al. (2010), where it is demonstrated that non core-shell internal particle morphologies can lead to "lensing" enhancements notably smaller than the core-shell solution.

Further, we reiterate that we measure the mass-weighted mean coating-to-core ratio. The actual population of particles sampled have a diversity of coating-to-core ratios. As has been established previously (Cappa et al., 2012;Fierce et al., 2016;Liu et al., 2017b), when the actual diversity of coating-to-core ratios across the population is accounted for smaller enhancements are obtained compared to that obtained assuming all particles have the mean coating-to-core ratio. For particularly diverse populations of particles, the population-weighted enhancement can be substantially less than that obtained assuming the mean coating-to-core ratio for all particles (Fierce et al., 2016). Thus, a single, or even a "few" calculations for some arbitrary particle morphologies would not, in our opinion, be especially informative. Instead, it would be necessary to perform calculations over a wide range of assumed particle populations and assumed internal morphologies to capture the likely true behavior. Such an exercise is well-beyond the scope of this manuscript, which is predominately observational, and further challenged by the lack of specific information regarding the internal particle morphology or the distribution of coating thicknesses across the particle population. Yet, we can give two examples here to illustrate the general conclusions we are likely to reach should we perform such calculations. Example 1: Assume two 130 nm diameter BC particles. One has a coating-to-core ratio of zero (bare BC) while the other has a coating-to-core ratio of 20. The mean coating to core ratio is 10. The core-shell Mie enhancement for the first particle is 1 while for the second is 2.45. So, the mean enhancement is 1.72, notably lower than an enhancement of 2.1 if the coating-to-core ratio for both particles were 10. While this enhancement is still quite a bit larger than 1, this illustrates the point that consideration of particle diversity generally leads to reductions in the enhancement. Example 2: Assume a log normal distribution of BC cores (mode diameter = 120 nm, width = 1.6). Assume also that the coating amount decreases with core size, consistent with condensational growth, with an (arbitrary) functional form illustrated in the following figure. Set the coating thickness distribution such that the mean coating-to-core ratio is 5, a "thick" value typical of our observations. The resulting coating-induced enhancement is 1.35 for the distribution, compared to a value of 1.79 if all particles had the same coating coating-to-core ratio. Again, a substantial reduction is observed for the more diverse (variable coating-to-core ratio) population compared to the constant coating case. Other functional forms for the coating distribution can be chosen, which would give different—albeit consistent—results, reflecting the particular nature of the assumed distribution; this includes distributions that yield even smaller ensemble-average enhancements for the same coating-to-core ratio. We can include the results of calculations of this sort in the manuscript. However, we do not feel that they are appropriate here given that we do not know the coating distribution. Additionally, we note that we have assumed a core-shell morphology here, which generally yields larger enhancements compared to more detailed calculations that account for internal particle morphology.

[Figure]

 Ultimately, given what the literature says about the importance of considering both internal morphology and of particle populations we believe it is a reasonable inference (and not a "hand waving argument") to conclude that our observations point to the lensing-induced absorption enhancement being "negligible for all conditions." We further believe that citation of the existing literature is preferred over the addition of calculations here. Nonetheless, as noted above, we have revised the statements in the abstract and throughout the manuscript as follows to make clearer that (i) we refer to results from our experiments and (ii) that the effect is "minor" rather than "negligible." Specifically, for the abstract we have revised to state: "The contribution of coating-induced enhancements (i.e., "lensing" effects) to BC absorption are shown minor for all of the burns despite some burns producing particles have large ensemble-average coating-to-core mass ratios."

Zhang, X., Mao, M., Yin, Y., and Wang, B.: Numerical Investigation on Absorption Enhancement of Black Carbon Aerosols Partially Coated With Nonabsorbing Organics, 
[revised manuscript text omitted]

| | 355, 405, 532, 1064 | 0.012, 0.0065, 0.0024, 0.0023 | | Photo-Acoustic Spectrometer | Alaskan & Indonesian Peat | central values reported here | (Sumlin et al., 2018) |
| | 600/400 ratio | | 0.04 | Water soluble organic carbon | Florida peat | Ratio between wavelengths reported | (Sengupta et al., 2018) |
| **Ambient** | 404 | 0.01 | 1.0-1.1 | Photo-Acoustic Spectrometer | Wild fire, near-source emission | Four Mile Canyon, Colorado | (Lack et al., 2012a) |
| | 470 | | 1.01 | Aethalometer | Biomass burning influenced | Beijing, China | (Yang et al., 2009) |
| | 400 | 0.112 | 2.9 | Filter transmission | Wood burning and biomass smoke aerosols | Savanna | (Kirchstetter et al., 2004) |
| | 532 | 0.0016-0.0019 | 0.029-0.031 | Photo-Acoustic Spectrometer | HULIS from biomass burning aerosols | Amazon basin | (Hoffer et al., 2006) |
| | Broadband | 0.05-0.07 | | Airborne lidar | Upwind of forest fires | Northern Canada | (Wandinger et al., 2002) |
| | Broadband | 0.07±0.03/ 0.04±0.01 | | White light optical particle counter | Open fire/ Smoldering phase | Urban Rehovot, Israel | (Adler et al., 2011) |
| | 405 | 0.037 | 0.79 or 1.22 | Photo-Acoustic Spectrometer | Residential biomass burning influenced | Fresno, CA | (Zhang et al., 2016) |
| | 405 | | 0.84 | Photo-Acoustic Spectrometer | Residential biomass burning influenced | Fresno, CA | (Cappa et al., 2019b) |
| | 405 | | 2.3 | Aethelometer | Biomass burning influenced | Guangzhou, China | (Qin et al., 2018) |
| | 365 | | 0.32 | Water soluble organic carbon | Plume intercept – closest point to fire | Western US | (Forrister et al., 2015) |
| | 365 | | 1.35 | Water soluble organic carbon | Regional biomass burning | SE US | (Washenfelder et al., 2015) |

| | | | | | | |
|---|---|---|---|---|---|---|
| 405 | 0.037 | | Water soluble organic carbon | Regional | Kanpur, India | (Shamjad et al., 2016) |
| 405 | | 0.7-1.3 | Water soluble organic carbon | Bonfire festival | Rehovot, Israel | (Bluvshtein et al., 2017) |
| 405 | | 0.6 | Methanol soluble organic carbon | Prescribed burn | NW US | (Xie et al., 2017) |
| 400, 600, 800 | 0.31, 0.26, 0.22 | | Electron loss | Asian outflow | Downwind of Asia | (Alexander et al., 2008) |
| 800/400 ratio | | 0.26 | Methanol soluble organic carbon | Ambient particles (ratio between wavelengths reported) | Athens, Georgia | (Phillips and Smith, 2017) |
| 400, 550, 700 | 0.112, 0.030, 0.001 | | Acetone treatment + attenuation | African biomass burning | Southern Africa | (Kirchstetter et al., 2004) |